# THE IDEATION–EXECUTION GAP: EXECUTION OUTCOMES OF LLM-GENERATED VERSUS HUMAN RESEARCH IDEAS

**Chenglei Si, Tatsunori Hashimoto, Diyi Yang**
Stanford University
{clsi,thashim,diyiy}@stanford.edu

## ABSTRACT

Large Language Models (LLMs) have shown promise in accelerating the scientific research pipeline. A key capability for this process is the ability to generate novel research ideas, and prior studies have found settings in which LLM-generated research ideas were judged as more novel than human-expert ideas. However, a good idea should not simply appear to be novel, it should also result in better research after being executed. To test whether AI-generated ideas lead to better research outcomes, we conduct an execution study by recruiting 43 expert researchers to execute randomly-assigned ideas, either written by experts or generated by an LLM. Each expert spent over 100 hours implementing the idea and wrote a 4-page short paper to document the experiments. All the executed projects are then reviewed blindly by expert NLP researchers. Comparing the review scores of the same ideas before and after execution, the scores of the LLM-generated ideas decrease significantly more than expert-written ideas on all evaluation metrics (novelty, excitement, effectiveness, and overall; $p < 0.05$), closing the gap between LLM and human ideas observed at the ideation stage. When comparing the aggregated review scores from the execution study, we even observe that for many metrics there is a flip in rankings where human ideas score higher than LLM ideas. This ideation-execution gap highlights the limitations of current LLMs in generating truly effective research ideas and the challenge of evaluating research ideas in the absence of execution outcomes. [1]

## 1 INTRODUCTION

LLMs have shown promise in various tasks in the scientific research pipeline, and most recently, they have been envisioned to power AI scientists that can autonomously make novel scientific discoveries. Recent efforts have built LLM-powered agentic systems to propose novel drug repurposing and treatment targets Gottweis et al. (2025); Ghareeb et al. (2025), develop new efficient matrix multiplication algorithms and optimal constructs for open mathematical problems (Novikov et al., 2025), and end-to-end produce full research papers on AI topics (Lu et al., 2024; Yamada et al., 2025).

Generating high-quality research ideas is the first step in these automated research pipelines, and the quality of LLM-generated ideas can decide the upper-bound of the final execution outcomes. Despite the importance of this ideation step, measuring the quality of LLM-generated research ideas is difficult, as it not only requires extensive domain expertise but also involves subjective taste. Prior attempts of evaluating LLM-generated research ideas mostly focus on the ideas themselves without considering the execution outcomes, with most evaluations relying on either LLM judges (Lu et al., 2024; Li et al., 2025; Feng et al., 2025) or small-scale human evaluation Baek et al. (2025); Wang et al. (2024a).

A recent large-scale human study examined AI-generated ideas in a randomized, blinded comparison to human experts (Si et al., 2025) and found that LLM ideas are judged as significantly more novel than human ideas with higher average scores across novelty, excitement, and expected effectiveness.

---

[1] All of our data are released at: https://github.com/NoviScl/AI-Researcher.

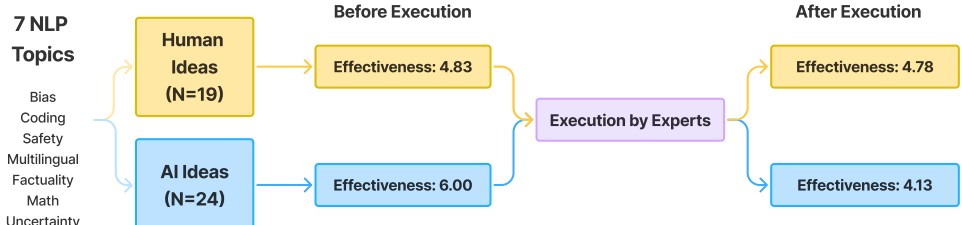

Figure 1: Study overview: we recruit $N = 43$ expert researchers to execute randomly assigned ideas from either the Human condition or the AI condition. Expert reviewers then blindly review all the executed projects. Despite the AI ideas being scored higher than human ideas before execution (e.g., their predicted effectiveness score of the ideas), their scores drop significantly more than human ideas after execution (e.g., their effectiveness score based on the experiment results).

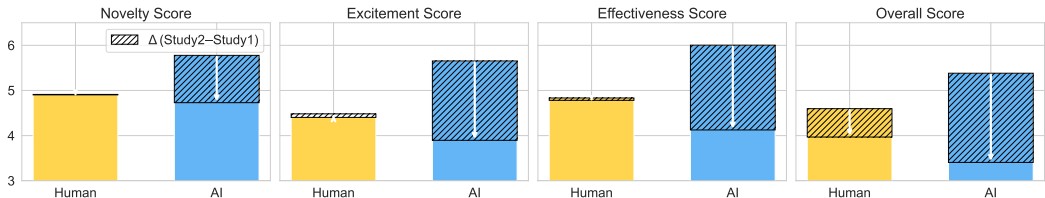

Figure 2: Average scores of AI ideas drop significantly more than Human ideas in the execution study across all the evaluation metrics. AI ideas score higher than Human ideas in the ideation evaluation (Study 1), and this difference in drops narrows their difference in the execution evaluation (Study 2). In fact, AI ideas score even lower than Human ideas in the execution evaluation, although this difference is not statistically significant (Table 4).

However, evaluating research ideas is difficult even for experts (Simsek et al., 2024), leaving open the question of whether these ideas would translate into better research outcomes.

There are several reasons why evaluation results on the ideas might not hold true when we execute them into actual projects. During execution, every single step has to be grounded in realistic execution constraints, which impose higher feasibility standards than the ideation stage. Moreover, objective metrics like feasibility and effectiveness are best judged via the actual execution outcomes rather than speculative judgment based on the ideas. However, execution is both resource- and time-consuming for many research domains. As a compromise, prior works are constrained to verify only one or two AI-generated ideas through execution experiments (Ghareeb et al., 2025; Gottweis et al., 2025), which makes it impossible to draw any statistically significant conclusions about LLMs' ideation capabilities.

Our work provides the first quantitative, large-scale study of AI ideas after execution by performing a large-scale execution study with a sufficient sample size to draw statistically significant conclusions about the post-execution quality of AI-generated ideas. We recruit $N = 43$ qualified participants and randomly assign each of them an NLP research idea from either a human researcher or an LLM agent based on `Claude-3.5-Sonnet`, a frontier LLM at the time when we started the study. Our execution study builds upon Si et al. (2025), and which enables us to use the collected ideas and their pre-execution ideation evaluation. Our execution participants spend an average of 103 hours executing the assigned idea and then submit the codebase and paper to document their experiments. All projects are then reviewed blindly by our recruited expert reviewers (Figure 1).

Comparing the review scores of these ideas from the previous ideation evaluation and our new execution evaluation, we observe the **ideation-execution gap of LLM-generated ideas**: LLM ideas score much lower in the execution evaluation as compared to the ideation evaluation. In contrast, human expert ideas only incur small drops from the ideation evaluation to the execution evaluation, significantly smaller than the ideation-execution gap of LLM-generated ideas ($p < 0.05$; Figure 2). Despite the significantly higher pre-execution ideation scores from LLMs, the huge ideation-execution

gap leads to a flip in the rankings between LLMs and humans. Our analysis further shows that reviewers consider more comprehensive factors in the execution evaluation, uncovering previously overlooked weaknesses of LLM ideas. These results add nuances to previous claims on LLMs generating better ideas than human experts, and perhaps more importantly, highlight the difficulty of evaluating research ideas in the absence of execution outcomes.

## 2 EXECUTION STUDY SETUP

In this section, we go over the high-level study design and lay out some specific rules for our execution process to ensure a controlled comparison between the Human and AI conditions.

### 2.1 HIGH-LEVEL STUDY DESIGN

Our goal is to compare the execution outcomes of LLM-generated research ideas and human experts' research ideas (Figure 1). We design a blinded, Randomized Controlled Trial (RCT) study: each of the $N = 43$ participants is randomly assigned to execute an anonymized idea originating from either the LLM or a human expert under identical instructions, with reviewers also blinded to the idea source. Each execution participant is given a three-month window to execute the assigned idea into a full project, including implementing and running all the proposed experiments and writing a 4-page paper in the specified format. By standardizing the execution and reviewing procedures, our RCT design ensures that statistically significant differences in execution outcomes can be causally attributed to whether the idea came from the AI or from a human. Our study design and hypotheses were pre-registered. [2]

| Topic | Human | AI |
|---|---|---|
| Bias | 3 | 4 |
| Coding | 3 | 3 |
| Safety | 2 | 4 |
| Multilingual | 3 | 4 |
| Factuality | 5 | 5 |
| Math | 1 | 1 |
| Uncertainty | 2 | 3 |
| Total | 19 | 24 |

Table 1: Topic distribution of the executed projects.

The LLM and human ideas used in our execution study are taken from a previous ideation study, spanning 7 different NLP topics (Table 1). These ideas were intentionally scoped to be feasible to execute within three months when they were collected in the ideation study. For the pre-execution evaluation, we use the ideation evaluation scores from the previous ideation study; for the post-execution evaluation, we recruit $N = 58$ expert reviewers to score the executed projects. These reviewers are drawn from a similar population as the ideation study, using review guidelines that closely match the ideation study wherever possible.

### 2.2 MINIMIZING CONFOUNDERS IN THE STUDY DESIGN

We establish some ground rules for the execution task. Our guiding principle is to minimize any changes to the assigned idea while maintaining blinding and random assignment across the two conditions.

**Random Idea Assignment**  When assigning ideas, we want execution participants to be able to work on ideas within their expertise, but at the same time, we want to avoid potential self-selection bias where participants only choose higher-quality ideas to work on. To balance this tension, we first ask for their preferred topics among our 7 candidate topics, and then we randomly assign either an LLM-generated idea or a human-written idea to them from their selected topics. This random assignment avoids potential selection biases and also helps randomize the participants' expertise levels across the two experiment conditions.

**Minimize Changes to Ideas**  Our goal is to evaluate the execution outcomes of the original ideas, and thus, we want to preserve the ideas as much as possible during execution. At the same time, we recognize cases where certain changes are necessary to make the ideas executable. To balance these, our policy is to disallow any substantial changes to the proposed methods from the original ideas,

---

[2] https://osf.io/ckxtp

| | Human Idea Executor (N = 19) | | AI Idea Executor (N = 24) | | Reviewer (N = 58) | |
|---|---|---|---|---|---|---|
| Metric | Mean | Median | Mean | Median | Mean | Median |
| No. of Papers | 15.3 | 5 | 14.3 | 11 | 13.6 | 11 |
| No. of Citations | 233.5 | 36 | 288.7 | 118 | 596.8 | 287 |
| Topic Familiarity (1-5) | 2.9 | 3 | 3.4 | 3 | 3.2 | 3 |
| Time | 112.6 hr | 120 hr | 93.7 hr | 104 hr | 52.5 min | 45 min |

Table 2: Research profiles of the execution and review participants and their efforts spent.

while allowing changes to the experimental details. We enforce this rule by asking all execution participants to explicitly note down all the changes they want to make. We manually review these proposed changes to ensure that they are focused on refining the experiment details, such as dataset choice and baseline selection, and we verified that these changes do not change the core method proposed in the assigned ideas (discussed in full detail in Section 5.1). Of all the assigned ideas, there is only one exception where the participant found the idea to be too vague, and they had to instantiate the method details with their own ideas to make the project feasible. We thus terminated that project and excluded it from the remaining study.

**Deliverables and Reviews** At the end of the execution, we ask all participants to submit the full codebase for reproducing all the experiments, adhering to our guidelines on writing detailed README instructions to reproduce the experiments, as well as a short paper of at least 4 pages in a standardized format. After the execution stage, we recruit a pool of qualified reviewers to do blind review of the executed projects, where both the codebase and the paper will be shown to our reviewers for their blind review. The review form is largely similar to conventional conference review forms and matches the style of the review form used in the previous ideation study, including metrics on novelty, excitement, soundness, effectiveness, and overall quality. Additionally, we also collect review scores on how faithful the execution is to the original idea and the codebase quality as control metrics. The full review form can be found in Appendix A.

## 3 EXECUTION AND REVIEWING PARTICIPANTS

A core premise of our study is to rely on highly qualified participants to execute the given ideas into the corresponding experiments, and to rely on blind reviewing from expert reviewers for a fair evaluation of all the executed outcomes. In this section, we describe the profiles of recruited experts and their efforts in this study.

### 3.1 EXPERT RECRUITMENT

We recruit our participants by posting recruitment messages on various social media platforms (e.g., Twitter/X and Slack), directly reaching out to qualified candidates, and advertising during in-person conferences. After basic profile screening, we onboarded a total of 66 participants for the execution task, among whom 43 completed the task in the end. Among the participants who did not complete the execution task, only one of them was because of the assigned idea was too vague and infeasible, while all the other cases were due to various personal reasons. Our 43 execution participants came from 7 different countries, including the US, Australia, India, the UK, Nepal, Singapore, and Canada.

For the execution task, we give each participant a three-month window to complete the task and compensate them for the total number of hours spent on the task ($20/h$) as well as a completion bonus ($600). Moreover, we reimburse all the compute costs incurred during the execution, such as inference API credits. Apart from the compensation, we also allow all execution participants to take ownership of the executed projects so that they can further develop the project after the study as potential paper submissions if they wish. This serves as an additional incentive for the participants, and in fact, multiple of them developed the executed projects into paper submissions after the study. For the review task, we recruited 58 highly qualified reviewers who collectively wrote 181 reviews, ensuring each project is reviewed by 4-5 different reviewers. Each reviewer is assigned 2-5 projects based on their preferred topics and is compensated $50 for each review they write.

| | Human Ideas (N=85) | | AI Ideas (N=96) | | |
|---|---|---|---|---|---|
| Metric | Mean | SD | Mean | SD | $p$-value |
| Novelty (1–10) | 4.93 | 1.61 | 4.73 | 1.75 | 0.21 |
| Excitement (1–10) | 4.52 | 1.84 | 3.90 | 1.75 | 0.01* |
| Effectiveness (1–10) | 4.84 | 2.11 | 4.12 | 1.94 | 0.01* |
| Soundness (1–10) | 5.38 | 1.72 | 4.73 | 1.82 | 0.01* |
| Overall (1–10) | 4.00 | 1.59 | 3.41 | 1.46 | 0.01* |
| Faithfulness (1–10) | 6.48 | 2.03 | 6.42 | 1.56 | 0.41 |
| Codebase Quality (1–5) | 3.58 | 0.94 | 3.58 | 0.89 | 0.52 |

Table 3: Results aggregated over all reviews for Human and AI conditions in the execution evaluation by treating each review as an independent data point. We perform two-sample one-sided t-tests to test whether the mean of the AI condition is smaller than the mean of the Human condition. We report the $p$-values with FDR correction to account for multiple hypothesis testing for the five main evaluation metrics in the first block, and report the raw $p$-values for the control metrics in the second block.

## 3.2 EXPERT QUALIFICATIONS AND EFFORTS

Finding qualified execution participants is crucial to avoid cases where the execution outcomes do not faithfully reflect the original idea's effectiveness due to poor execution. We took measures such as collecting candidates' profiles and conducting screening interviews to find participants who are both qualified and highly motivated and committed to the task. All of our execution participants have substantial prior research background, with an average of 15.3 and 14.3 papers on their Google Scholar profiles for the Human and AI conditions, respectively (first block of Table 2). We also collected their self-reported familiarity with the assigned topic, for which they indicated moderately high familiarity (2.9 and 3.4 on a 1-5 scale for Human and AI conditions; Table 2). Apart from being highly qualified, our execution participants spent an average of 112.6 and 93.7 hours executing human and AI ideas, indicating substantial effort. This level of variation in expertise and time spent between the human and AI conditions is expected, given our relatively small sample size. Similarly, our expert reviewers have extensive research experience (596.8 average citations) and are generally familiar with the reviewed topics (3.2 out of 5); and spent an average of 52.5 minutes on each review (last two columns of Table 2). Our pool of participants is diverse and comes from 40 different institutions across the world. We present the institutions that our execution and reviewing participants come from in Appendix B.

## 4 QUANTITATIVE RESULTS

In this section, we summarize our main quantitative results from the execution study. Our main goal is to compare AI ideas and human ideas, and we will compare them both in terms of the scores from the execution study, as well as the differences compared to the ideation evaluation.

## 4.1 COMPARING HUMAN AND AI IDEAS IN THE EXECUTION STUDY

A natural outcome measure is the review scores from the execution study. We compare the scores of human and AI ideas in the execution evaluation in Table 3, where we treat each review as an independent data point and aggregate the scores from all reviews. Human ideas and AI ideas score similarly on our control metrics – faithfulness and codebase quality, indicating similar execution quality and that ideas from both conditions are executed faithfully. When treating each review as an independent data point (sample size $N = 181$) and performing one-sided t-tests, human ideas score significantly higher than AI ideas on excitement, effectiveness, soundness, and the overall score, but not novelty. However, when we compare human and AI ideas by treating the average score of each idea as an independent data point (sample size $N = 43$), the difference between human and AI ideas' execution score is not significant on any metric (second block of Table 4). Given this mixed evidence, we do not have sufficient statistical power to directly confirm our pre-registered hypothesis that human ideas differ significantly from AI ideas in the execution scores. Additionally,

|                                | Novelty | Excitement | Effectiveness | Overall |
|--------------------------------|---------|------------|---------------|---------|
| Human Condition Ideation Score | 4.912   | 4.404      | 4.833         | 4.596   |
| AI Condition Ideation Score    | 5.778   | 5.653      | 6.003         | 5.382   |
| $p$-value (FDR)                | 0.035*  | 0.004**    | 0.001**       | 0.035*  |
| Human Condition Execution Score| 4.903   | 4.482      | 4.782         | 3.968   |
| AI Condition Execution Score   | 4.729   | 3.896      | 4.125         | 3.406   |
| $p$-value (FDR)                | 0.586   | 0.175      | 0.266         | 0.175   |

Table 4: Comparison of mean ideation vs. execution scores for human and AI conditions. We treat the average score of each idea as an independent data point, so the sample size is $N = 19$ for the human condition and $N = 24$ for the AI condition. For the $p$-values, we perform two-sided t-tests with FDR correction. $*$ means $p < 0.05$, $**$ means $p < 0.01$.

|                    | Novelty | Excitement | Effectiveness | Overall |
|--------------------|---------|------------|---------------|---------|
| Human Ideas Gap    | $-0.010$ | 0.078     | $-0.052$      | $-0.628$ |
| AI Ideas Gap       | $-1.049$ | $-1.760$  | $-1.879$      | $-1.976$ |
| $\Delta$ (Human–AI)| 1.039   | 1.835      | 1.827         | 1.348   |
| $p$-value (FDR)    | 0.025*  | 0.001**    | 0.003**       | 0.004** |

Table 5: Comparison of the gaps between the execution evaluation and the ideation evaluation scores for human and AI ideas. Negative gaps indicate a score decrease after execution. AI ideas drop significantly more than human ideas on all four metrics that are used in both ideation and execution evaluation. $*$ means $p < 0.05$, $**$ means $p < 0.01$. All $p$-values are adjusted with FDR correction.

we plot the distribution of all projects' scores in the execution evaluation in Appendix C and the reviewer agreement in Appendix D, where we show that the reviewer agreement is generally high.

## 4.2 MEASURING THE IDEATION-EXECUTION GAP

Directly comparing average human and AI idea scores is difficult due to the high heterogenity in idea quality. However, the design of our study provides a natural way to remove this variance, instead of comparing the direct ratings, we can compare the difference between pre- and post- execution scores.

This *ideation-execution gap* controls for the heterogenity in idea quality, and results in clear statistical signals even with 43 projects. Concretely, we focus on the four metrics that are used in both evaluations: novelty, excitement, effectiveness, and the overall score. To define the metrics, we take the difference in score between the execution evaluation and the ideation evaluation (so negative gaps would mean the scores decreased after execution). Figure 2 and Table 5 show that AI ideas' scores drop significantly more than human ideas in the execution evaluation across all four metrics. For example, human ideas mostly retain the same novelty, excitement, and effectiveness scores after execution ($-0.010$, $+0.078$, and $-0.052$) while AI ideas' scores decrease by $1.049$, $1.760$, and $1.879$ after execution, on a 1-10 scale.

In Table 4, we present the full scores of all projects before and after the execution. Prior to the execution, AI ideas score significantly higher than human ideas on all four metrics. However, due to the bigger ideation-execution gaps of AI ideas, the gap between human and AI ideas in the execution evaluation shrinks. In fact, we see a case where the rankings flip, and AI ideas score below human ideas after execution on all metrics (e.g., 3.90 v.s. 4.48 on excitement and 4.13 v.s. 4.78 on effectiveness). However, we note that such differences are not statistically significant due to the small sample size when treating each idea as an independent data point ($N = 43$). We further compute the correlation between the ideation scores and execution scores in Appendix E, where the correlation is weak in most cases; and for AI ideas, there is even a moderately negative correlation on the excitement score.

## 5 ANALYZING THE IDEATION-EXECUTION GAP

In this section, we dive deeper into the executed projects and perform both quantitative and qualitative analyses to understand why there exist ideation-execution gaps when we execute ideas into projects.

### 5.1 CHANGES MADE TO THE IDEAS MAINLY FOCUS ON EXPERIMENT DETAILS

We begin by analyzing the types of changes made to the ideas by our execution participants. We manually annotate all the changes to construct a taxonomy of the types of changes. We show the counts of all types of changes made to human and AI ideas in Table 6, where we see that human ideas and AI ideas involve an average of 2.9 and 3.1 changes, respectively. This indicates that only a moderate number of changes are made to both human and AI ideas. Moreover, all of the changes focus on experiment details rather than altering any algorithms proposed in the original ideas. Examples of these changes include switching to benchmarks that are appropriate for the given tasks, updating the backbone models to more recent ones, adding more comprehensive evaluation metrics, specifying any missing hyper-parameters and prompt details, adding stronger baselines, and adding more analysis or ablation studies. We present examples of each type of change in Appendix F. These changes generally preserve the original ideas and improve the experiment design.

However, we do note one exception: when we prompted both humans and AI to generate ideas that can be executed within three months, humans are better at scoping the experiments to be more feasible. The most common example is that AI-generated ideas like to propose human evaluations by recruiting experts or native speakers to annotate a large set of model predictions, which are always changed by the executors to save cost and time. For example, for the AI idea "Sociolinguistic Role-Play Prompting", the idea originally proposed to conduct a human study by recruiting native speakers of different languages and cultural experts to rate model generated outputs, which is one of the major reasons for high excitement scores of this idea during ideation evaluation ("the analysis with real native speakers of different languages could be a major contribution of this work"). However, this human study was

| Type of Change | Human | AI |
|---|---|---|
| Sample Size | 19 | 24 |
| Dataset Change | 11 | 18 |
| Metric Change | 5 | 13 |
| Human –¿ Auto Eval | 0 | 6 |
| Model Change | 10 | 12 |
| Hyper-Parameters | 7 | 7 |
| Baseline Change | 6 | 4 |
| Analysis Change | 4 | 7 |
| Prompt Details | 13 | 7 |
| Total | 56 | 74 |
| Average | 2.9 | 3.1 |

Table 6: Distribution of changes made to the original ideas during execution.

changed to using LLM-as-a-judge for automatic evaluation, which became a weakness as noted by the reviewer: "Without manual evaluation, it is hard to gauge whether the outputs really improve in terms of cultural adaptability or there are other data artifacts that LLM judges rely on when making their preference decisions. " We manually annotated all ideas to count such cases, and out of all the 43 ideas, 6 AI ideas had such changes where the proposed human evaluation was changed to automatic evaluation. However, removing all these 6 ideas does not change any of our previous conclusions, and AI ideas still incur significantly larger ideation-execution gaps than human ideas (full results in Appendix G). We thus conclude that these changes only explain a small fraction of the ideation-execution gap.

### 5.2 EXECUTION EVALUATION CONSIDERS MORE FACTORS THAN IDEATION EVALUATION

Given that the changes made to the ideas only explain a small fraction of the ideation-execution gap, we then turn to the review differences between the ideation and execution evaluations. Specifically, we hypothesize that reviewers are focusing on very different aspects between the ideation and the execution evaluation, which results in the ideation-execution gaps. For each idea, we analyze the free-form reviewer comments and summarize the main points. We then manually categorize all the reviewer comments into one of the following categories: (1) novelty and motivation of the idea; (2) significance or impact of the idea; (3) technical flaws of the proposed method; (4) experiment design (issues with model selection, dataset, metrics, or evaluation methods); (5) baseline comparison; (6) ablation and analysis; (7) practical feasibility and resource requirements; (8) empirical performance;

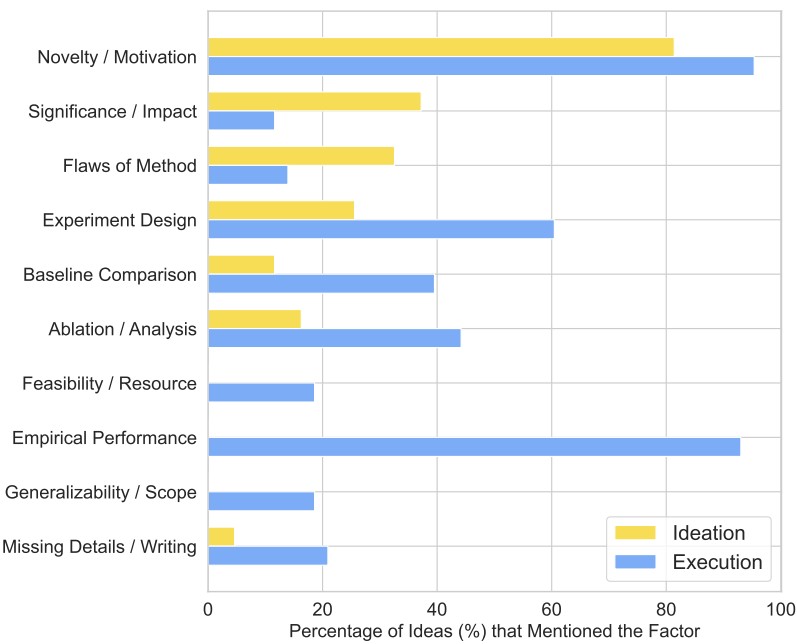

Figure 3: Comparison of the factors mentioned in the reviewer rationales in the ideation (yellow bars) and the execution evaluation (blue bars). The y-axis indicates the percentage of ideas where the reviews mention the corresponding factor. Each idea's reviews could mention multiple factors. In execution evaluation, reviewers consider more factors, especially the rigor and soundness of experiments and the empirical performance.

(9) generalizability and scope; and (10) missing details or bad writing. We present the quantitative breakdown in Figure 3 and summarize several important trends below.

**First, execution evaluation always considers empirical experiment outcomes, while the ideation evaluation often speculates.** Since there was no empirical performance presented during the ideation evaluation, the ideation evaluation is often conditioned on the assumption that the method would be effective. For example, the reviewer noted "*if the experiments show significant improvements to direct using external knowledge/self-reflection and the work provides insightful analysis on why, I believe it is exciting enough to get published*" for the AI-generated idea "Adaptive Confidence-Guided Prompting" during ideation evaluation. Similarly, the reviewer noted "*This is pretty hard to predict. Assuming the experiments are successful and thorough, it would be a solid paper worthy of acceptance at any conference. However, this is entirely dependent on how the experiments turn out. It is entirely possible that the proposed method is ineffective, and we don't learn anything substantial from it, in which case a paper might not even exist.*" for the AI idea "Conceptual Pivot Prompting for Bias Mitigation". In contrast, almost all the reviewers would consider empirical performance in the execution evaluation, and their scores often heavily depend on this factor. For example, the reviewer noted "*the method does not show marked improvements over a basic empathy prompting approach*" for the AI-generated idea "Empathetic Cascading Networks" to justify a low overall score.

**Second, execution evaluation poses a higher standard for the rigor of empirical evaluation.** While the ideation evaluation rarely focuses on the experiment design details, the execution evaluation emphasizes this much more. For example, the reviewer detected the issue with evaluation metrics ("*not using the same metrics as other works to compare the efficacy of this method*") for the AI idea "Temporal Bias Decay Simulation", which was previously overlooked in the ideation evaluation without observing the executed experiments. Moreover, empirical experiments inspire reviewers to notice additional weaknesses of the idea and the experiment design, such as missing baselines and ablations, high resource requirements, and poor generalizability, which are almost entirely overlooked during ideation evaluation. For example, one reviewer commented "*lacks comparison with*

*previous work: the method is only compared with the simplest baselines despite well-acknowledged benchmarks*" for the AI-generated idea "Contrastive Semantic Pivot Prompting" to criticize missing baselines; one reviewer noted "*the experiments should not just be numbers, but also include discussion of why ACGP actually produced the results provided*" for the AI-generated idea "Adaptive Confidence-Guided Prompting" to request more analysis; and one reviewer commented "*the method is also very computationally expensive*" for the AI idea "Adaptive Contextual Pruning" to criticize the resource consumption.

Taken together, the reviewers are considering more factors during the execution evaluation and grounding their judgment on the executed experiments, therefore uncovering more weaknesses of the ideas that were previously overlooked during ideation evaluation. For additional analysis, we also analyze the correlation between different evaluation metrics in Appendix H, where we find the overall score correlates highly with all the other breakdown metrics in the execution evaluation.

We showcase two randomly selected examples from our execution study in Appendix I. For each example, we present the original idea proposal and the executed paper, as well as the corresponding review scores in the ideation and execution evaluations.

## 6    RELATED WORK

**Research Idea Generation and Evaluation** Recent works have been exploring methods to generate novel research ideas or hypotheses with LLMs. Most of them focus on building better scaffolds, for example by integrating retrieval (Wang et al., 2024a; Li et al., 2025; Wang et al., 2024b), iterative feedback and revision (Yang et al., 2024; Hu et al., 2024), and multi-agent collaboration (Su et al., 2025). Some also attempted to train specialized idea generation models by finetuning open LLMs (Weng et al., 2025; O'Neill et al., 2025). These works typically use various forms of LLM judges for the automatic evaluation of the generated ideas, sometimes validated by small-scale human evaluation (Wang et al., 2024a). Several works also proposed more effective ways to leverage LLMs for automated idea evaluation, for example by incorporating the graph structure of research ideas (Feng et al., 2025) or training specialized reviewer models (Zhu et al., 2025). Among them, Si et al. (2025) is an exception where they conducted the first large-scale expert evaluation of LLM-generated ideas and revealed limitations of LLMs as automatic judges. However, all of these works consider the setting of idea evaluation without the execution outcomes, which, as we show, could deviate from the real execution outcomes.

**LLMs as AI Scientists** Relatedly, a growing body of work has been building LLM-based AI Scientists that could automate the entire scientific research pipeline including both ideation and execution. For example, various types of AI Scientist agents have been developed (Lu et al., 2024; Schmidgall et al., 2025; Yamada et al., 2025) for automating AI research, in which case the execution is realized through code generation. In other scientific domains like biology and chemistry, AI Scientists have been built to perform literature review and generate novel hypotheses and experiment plans, which are then validated in wet lab studies (Gottweis et al., 2025; Ghareeb et al., 2025). In verifiable domains like coding and math, LLMs have been used to power evolutionary searches by iteratively proposing new ideas based on prior execution feedback (Novikov et al., 2025; Zhang et al., 2025). The evaluation of such AI Scientist systems has been a major challenge, especially for open-ended research without objective success metrics. While existing works rely on automatic judges and occasionally submit cherry-picked examples to peer reviews, we conduct the first large-scale execution evaluation by recruiting experts for the execution to obtain reliable evaluations of the ideas.

## 7    DISCUSSION

In conclusion, we have conducted the first large-scale execution evaluation to assess the execution outcomes of LLM-generated research ideas in comparison with expert ideas. We find large ideation-execution gaps in LLM-generated ideas and advocate that future work should also take into consideration the execution outcomes when evaluating AI-generated ideas. We recognize that our study is still limited in the sample size and ideation scope despite our best efforts, and future work could explore the use of automatic coding agents to scale up the idea execution. We discuss various additional limitations and concrete future directions below.

## 7.1 LIMITATIONS

**Idea Scope** For this execution study, we directly reused the human and AI-generated ideas from our prior ideation study (Si et al., 2025), in which the scope was intentionally constrained to focus on novel prompting techniques. This constraint was necessary to ensure that the ideas were feasible to implement within a limited timeframe and compute budget, and accessible to a broader pool of executors with varying technical backgrounds. However, this focus may limit the generalizability of our findings to other types of research ideas – such as those requiring complex modeling and large-scale data collection or training. Future studies could expand the idea space to include a more diverse range of AI and human-generated proposals to better assess performance across different topics and complexity, and examine whether our findings transfer to other domains outside of AI research.

**Sample Size** We have made our best attempt to recruit as many qualified executors as possible given the constraints of time, budget, and available participant pool. Despite this effort, our final sample size of $N = 43$ executed projects remains modest. While sufficient for detecting meaningful differences in the ideation-execution gaps, the sample size may reduce statistical power for other more granular analyses. Moreover, with a limited number of projects per condition, results may be more susceptible to individual executor variability. Future replications with larger and more diverse executor pools would strengthen the robustness and external validity of our findings.

## 7.2 FUTURE WORK

**Automated Execution** In this study, we recruited highly qualified human researchers to execute both human- and AI-generated ideas to ensure high-quality and faithful implementations. While there has been rapid progress in building coding agents for automating ML engineering tasks (Chan et al., 2025; Liu et al., 2025) and replicating published research papers (Starace et al., 2025; Hua et al., 2025), current systems still suffer from low reliability and poor generalization to complex, open-ended tasks like research execution. A promising direction for future work is to develop more capable research agents that can autonomously implement research ideas at near-human levels of quality. Such agents could greatly improve the scalability of large-scale idea evaluation and experimentation by reducing the dependence on expert human labor.

**Proxy Reward Models** Executing research ideas is resource-intensive, requiring substantial human effort, time, and computational resources. One avenue to mitigate this cost is the development of proxy reward models – predictive models that can estimate the likely effectiveness an idea without requiring full implementation. These models could be trained on historical execution outcomes, for example, from papers with known empirical results (Wen et al., 2025); or leverage simulations of the execution environments. If successful, such models could serve various purposes, such as rapidly ranking and filtering generated ideas, and acting as reward functions in reinforcement learning pipelines for idea generation.

**Execution Feedback Loop** Another compelling direction is to build closed feedback loops where the outcomes of executed experiments inform iterative idea improvement. This could be achieved through training-free methods such as evolutionary search or self-refinement mechanisms, where generated ideas are mutated and selected based on past execution performance. Alternatively, execution feedback can be directly used in learning-based frameworks—for example, by using empirical outcomes as reward signals in reinforcement learning or fine-tuning procedures. Such feedback-driven pipelines would bring idea generation closer to autonomous scientific discovery.

## ETHICS STATEMENT

**Intellectual Property** We made an agreement with the experts who contributed the human ideas that if their ideas are executed by others, the executors can take ownership of the executed projects. And we allow executors to further develop the executed idea beyond our study, for example if they wish to turn it into a conference or workshop submission. This policy is designed to incentivize high-quality execution by granting executors full authorship rights over their implementations.

**Reviewing and Publication Policy** For evaluation purposes, we rely on a recruited pool of compensated reviewers rather than submitting the executed projects to external peer-reviewed conferences

or workshops. This choice serves two goals: reducing additional burden on the volunteer-based academic reviewing system, and enabling a more controlled and balanced evaluation process, wherein we can ensure each reviewer assesses a fair mix of AI- and human-generated ideas. While we do not prohibit executors from submitting their projects to external venues, we require that they explicitly disclose the original source of the idea—whether AI- or human-generated—if they choose to publish the work on public platforms (e.g., arXiv) or submit it to peer-reviewed venues. We also debriefed participants with this policy, also notifying them that correctness and quality of work that they post publically or submit is their reponsbility.

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

# A    FULL REVIEW FORM

We use the following review form to elicit reviews from all expert reviewers. Reviewers have one week of time to finish each review.

**1. Name**

**2. Institution**

**3. Email**

**4. Consent**

**5. Honor Code**: I confirm that I will not use ChatGPT, Claude, Gemini, or any other AI tools when writing my reviews.

**6. Familiarity**: Before reviewing the idea, please indicate how familiar you are with the given topic on a scale of 1 - 5 (this is just for us to understand potential confounders).

1. You have never read about this topic before
2. You have read at least one paper on this topic
3. You have read multiple papers on this topic but have not published any paper on it
4. You have co-authored at least one paper on this topic
5. You have co-authored multiple papers on this topic or have published at least one first-author paper on this topic

**7. Experience**: Have you reviewed for major NLP or AI conferences before (e.g., *ACL, COLING, NeurIPS, ICLR, ICML, AAAI)?

**8. Executed Paper**

**9. Executed Codebase**

**10. Novelty Score**: Whether the proposed idea in the paper is creative and different from existing works on the topic, and brings fresh insights. You are encouraged to search for related works online. You can consider all papers that have been accepted and published prior to December 2024 as existing work when judging the novelty.

1. Not novel at all: The idea is essentially identical to many existing papers, with no meaningful differences. It does not introduce any new insights or perspectives.
2. 
3. Mostly not novel: The idea is very similar to existing work, with only minor variations. You can easily find multiple papers presenting nearly the same concept.
4. 
5. Somewhat novel: The idea has some differences from existing work, but the variations are very incremental rather than substantial. It might refine or extend previous ideas but lacks enough originality to justify a new paper on its own.
6. Reasonably novel: The idea introduces notable differences compared to prior work and likely has enough originality to justify a new paper. However, it still builds significantly on existing ideas rather than breaking new ground.
7. 
8. Clearly novel: The idea presents major differences from all known existing works. It introduces fresh insights or approaches that significantly advance the topic in a meaningful way.
9. 
10. Very novel: The idea is highly original and substantially different from all existing work. It offers a groundbreaking, clever, or unexpected perspective that is both innovative and impactful.

11. **Novelty Rationale**: Short justification for your score. If you give a low score, you should specify similar related works. (Your rationale should be at least 2-3 sentences.)

12. **Excitement Score**: How exciting is this paper? Do you expect the idea or results to be very impactful? Would this work change the field and be very influential?

1. Poor: You cannot identify the contributions of this work, or it's not interesting at all and you would fight to have it rejected at any major AI conference.
2. 
3. Mediocre: This work makes marginal contributions and is very incremental.
4. 
5. Leaning negative: It has interesting bits but overall not exciting enough.
6. Learning positive: It is exciting enough to be accepted at a major AI conference, but still has some weaknesses or somewhat incremental.
7. 
8. Exciting: It would deepen the community's understanding or make major progress in this research direction.
9. 
10. Transformative: It would change the research field profoundly and worth a best paper award at major AI conferences.

13. **Excitement Rationale**: Short justification for your score. (Your rationale should be at least 2-3 sentences.)

14. **Soundness Score**: Is this paper technically sound? Are all the methodological details technically correct? Are the experiments well-designed to verify the proposed method or hypotheses? Are they using appropriate datasets, metrics, and baselines? Overall, is this project well-executed?

1. Fundamentally flawed: The paper has major technical errors, incorrect methodologies, or logical inconsistencies that invalidate its conclusions. Experiments, if present, are deeply flawed or missing.
2. 
3. Seriously unsound: It has significant methodological flaws or missing technical details make it difficult to trust the findings. Experiments are poorly designed, use inappropriate datasets or baselines, or do not sufficiently verify the claims.
4. 
5. Somewhat unsound: The methodology is mostly reasonable, but there are a few notable gaps in correctness, experimental design, or justification.
6. Reasonably sound: The methodology is generally correct and the experiments are reasonable, but some minor technical choices, suboptimal experimental design, or missing details could be further improved.
7. 
8. Clearly sound: The paper is well-executed with a solid methodology, proper experimental design, and appropriate datasets and baselines. Any issues are minor and do not significantly affect the conclusions.
9. 
10. Technically flawless: The paper is exceptionally well-executed, with rigorous methodology, well-designed experiments, and strong justifications for all methodological choices. No technical flaws or weaknesses.

15. **Soundness Rationale**: Short justification for your score. (Your rationale should be at least 2-3 sentences.)

16. **Effectiveness Score**: Now focus on the experiment results. Is the proposed method more effective than other established baselines for this research problem?

1. Not effective at all: The proposed method performs significantly worse than all existing baselines.

2.

3. Mostly ineffective: The proposed method is mostly on par with existing baselines. No evidence suggests any significant improvement.

4.

5. Mixed results: The method provides mixed results. It works better than baselines on some datasets or metrics, but not consistently across all of them. The gains tend to be very small and not significant.

6. Reasonably effective: The method shows noticeable improvements over baselines on some datasets and metrics and is on par with baselines in the other settings. There may be some caveats or trade-offs between different datasets or metrics.

7.

8. Clearly effective: The method demonstrates strong and consistent improvements over baselines across multiple datasets or metrics. The results are convincing and well-supported.

9.

10. Extremely effective: The method significantly outperforms all relevant baselines in a substantial and meaningful way. The improvements are large, robust, and generalizable across different settings.

**17. Effectiveness Rationale**: Short justification for your score. (Your rationale should be at least 2-3 sentences.)

**18. Codebase Quality**: Take a look at the provided codebase. Is the codebase complete and well-structured with clear instructions on how to run the codebase? How easy do you expect it to be for someone else to reproduce the experiments in the paper with this given codebase?

1. The codebase is clearly incomplete and problematic. They would not be able to reproduce the results here no matter how hard they tried.

2. The codebase is not well-documented despite having all the code. They would be hard pressed to reproduce the results.

3. The codebase is reasonably clean and documented. They could reproduce the results with enough effort.

4. The codebase is well-structured and documented. They could mostly reproduce the experiments by following the documentation.

5. The codebase is very well-structured and documented. They could easily reproduce all the experiments by following the documentation.

**19. Overall Score**: Apart from the above, you should also give an overall score for the paper on a scale of 1 - 10 as defined below. Note that you should treat this paper as a short paper submission similar to the 4-page short paper track at *ACL (meaning that you should calibrate your expectation for the amount of experiments and analysis).

1. Critically flawed, trivial, or wrong

2. Strong rejection for major AI conferences

3. Clear rejection for major AI conferences

4. Ok but not good enough, rejection for major AI conferences

5. Decent idea but has some weaknesses or not exciting enough, marginally below the acceptance threshold of major AI conferences

6. Marginally above the acceptance threshold of major AI conferences

7. Good idea, would be accepted by major AI conferences

8. Top 50% of all published ideas on this topic at major AI conferences, clear accept

9. Top 15% of all published ideas on this topic at major AI conferences, strong accept

10. Top 5% of all published ideas on this topic at major AI conferences, will be a seminal paper

**20. Overall Rationale**: You should also provide a rationale for your overall score. (Your rationale should be at least 2-3 sentences.)

**21. Faithfulness Score**: Next, we present to you an outline for the core idea and experiments of the paper. Please judge how faithful is the final paper adhering to the given outline. [Original idea outline provided.]

1. Not faithful at all: The final paper completely deviates from the given outline, introducing a different core idea and experimental setup. Key components are missing or drastically altered.

2.

3. Mostly unfaithful: The paper retains some elements from the outline but introduces major changes to the core idea, making it significantly different from the original plan.

4.

5. Somewhat faithful: The paper follows the general idea in the outline, but there are notable deviations in key aspects of the methodology and experiment design.

6. Reasonably faithful: The core idea remains intact, and most experimental designs match the outline, though there are some notable changes to the implementation details and experiment setups.

7.

8. Clearly faithful: The paper closely follows the core idea in the given outline, with only minor modifications or refinements that do not alter the key ideas.

9.

10. Perfectly faithful: The final paper adheres precisely to the outline, including the key ideas and experimental designs without any significant changes.

**22. Faithfulness Rationale**: You should also provide a rationale for your faithfulness score. You are encouraged to reference specific sections or details in the paper and the outline. (Your rationale should be at least 2-3 sentences.)

**23. Confidence**: Additionally, we ask for your confidence in your review on a scale of 1 to 5 defined as following. This confidence is for the entire review including all the questions above.

1. Your evaluation is an educated guess.

2. You are willing to defend the evaluation, but it is quite likely that you did not understand central parts of the paper.

3. You are fairly confident that the evaluation is correct.

4. You are confident but not absolutely certain that the evaluation is correct.

5. You are absolutely certain that the evaluation is correct and very familiar with the relevant literature.

**24. Time**: How many minutes did you spend on this task? (Just provide an integer number.)

## B  Institutions of Execution and Reviewing Participants

| Institution | Count |
|---|---|
| University of North Texas | 4 |
| University of Southern California | 3 |
| Columbia University | 2 |
| IIIT Hyderabad | 2 |
| Northwestern University | 2 |
| University of Alberta | 2 |
| University of Maryland | 2 |
| University of Memphis | 2 |
| University of North Carolina | 2 |
| Carnegie Mellon University | 1 |
| Cornell University | 1 |
| Georgia Institute of Technology | 1 |
| IIT Madras | 1 |
| Kathmandu University | 1 |
| Massachusetts Institute of Technology | 1 |
| Microsoft Research | 1 |
| National University of Singapore | 1 |
| New York University | 1 |
| Nanyang Technological University | 1 |
| Penn State University | 1 |
| Stanford University | 1 |
| UC Berkeley | 1 |
| University of Bologna | 1 |
| University of Colorado | 1 |
| University of Illinois Urbana-Champaign | 1 |
| University of Melbourne | 1 |
| University of Oxford | 1 |
| University of Texas at Austin | 1 |
| University of Toronto | 1 |
| University of Notre Dame | 1 |
| Virginia Tech | 1 |

Table 7: Institutions of the 43 execution participants.

| Institution | Count |
|---|---|
| Stanford University | 20 |
| Georgia Institute of Technology | 4 |
| Princeton University | 4 |
| University of Southern California | 4 |
| University of Washington | 4 |
| University of Texas at Austin | 3 |
| Johns Hopkins University | 2 |
| Carnegie Mellon University | 2 |
| University of Illinois Urbana-Champaign | 2 |
| University of Maryland | 2 |
| Northeastern University | 2 |
| Yale University | 1 |
| New York University | 1 |
| Columbia University | 1 |
| University of Pennsylvania | 1 |
| University of Chicago | 1 |
| UC Berkeley | 1 |
| National University of Singapore | 1 |
| UC Santa Barbara | 1 |
| University of Michigan | 1 |

Table 8: Institutions of the 58 reviewer participants.

## C  PLOTS OF SCORE DISTRIBUTIONS

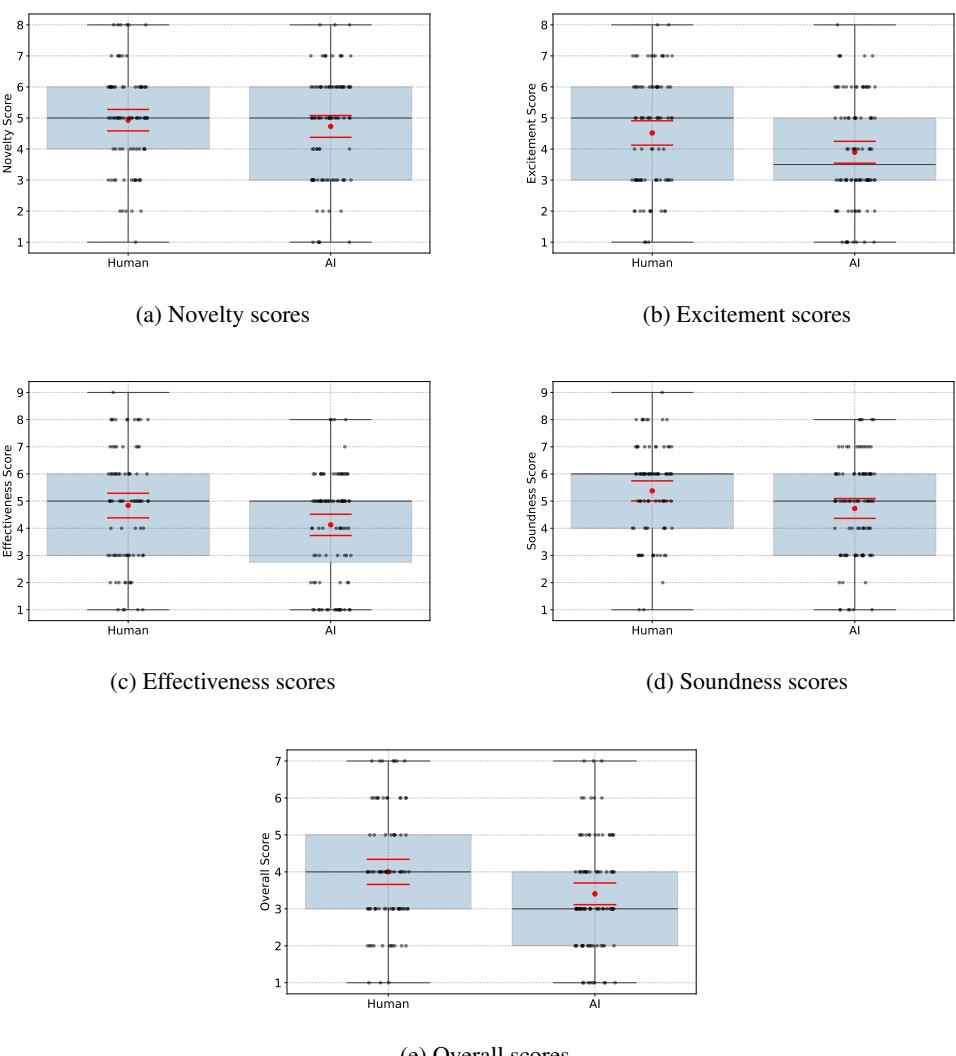

(a) Novelty scores

(b) Excitement scores

(c) Effectiveness scores

(d) Soundness scores

(e) Overall scores

Figure 4: Boxplots of all evaluation metrics across Human and AI conditions in the execution evaluation. Each point corresponds to one review. The red dots and red bars indicate the mean score and 95% confidence intervals.

# D    REVIEWER AGREEMENT

|  | Consistency |
|---|---|
| Random | 50.0 |
| NeurIPS'21 | 66.0 |
| ICLR'24 | 71.9 |
| Ours - Novelty | 67.0 |
| Ours - Excitement | 68.3 |
| Ours - Soundness | 59.0 |
| Ours - Effectiveness | 84.3 |
| Ours - Overall | 70.5 |

Table 9: Review score consistency among human reviewers.

Following the consistency metric used in Si et al. (2025), we randomly split reviewers of each paper into two halves, use one half to rank the top and bottom 25% of all ideas, and then measure agreement with the held-out set of reviewers. We used data from the NeurIPS 2021 reviewer consistency experiment and OpenReview data from ICLR 2024 to compute the baselines. As shown in Table 9, the reviewer agreement in our execution study is generally comparable to NeurIPS and ICLR across all metrics. Notably, the agreement on effectiveness is especially high, which is expected given that this is one of the more objective metrics that can be derived based on the actual experiment results.

# E    CORRELATION BETWEEN IDEATION AND EXECUTION SCORES

|              |                    | Novelty | Excitement | Effectiveness | Overall |
|--------------|--------------------|---------|------------|---------------|---------|
| AI ideas     | Pearson's $r$      | -0.019  | -0.321     | 0.172         | -0.092  |
|              | Spearman's $\rho$  | 0.077   | -0.386     | 0.110         | -0.092  |
| Human ideas  | Pearson's $r$      | -0.084  | 0.205      | 0.022         | 0.158   |
|              | Spearman's $\rho$  | -0.148  | 0.183      | -0.092        | 0.124   |

Table 10: Correlation between ideation scores and execution scores for AI ideas and human ideas, respectively. We present both the Pearson's correlation coefficient ($r$) and Spearman's rank correlation ($\rho$). The correlation is weak in most cases, and for AI ideas, there is even a moderately negative correlation on the excitement score.

## F   CHANGES MADE TO THE IDEAS

We present representative examples for each type of change that the executors made to the original ideas.

- Dataset Change: This refers to changes made to any datasets involved in the experiments. For example, the AI-generated idea "Contrastive Semantic Pivot Prompting" mentioned experiments on "Ethical dilemmas from the Moral Scenarios dataset", which was removed by the participant because this dataset does not exist. In another example, the AI-generated idea "Sociolinguistic Role-Play Prompting" proposed experiments on OpenSubtitles and XNLI, which were both removed because they don't contain the sociolinguistic metadata necessary for the proposed experiments. In the AI-generate idea "Adaptive Semantic Masking", the executor added more datasets, including Jailbreak-Bench and DAN-Forbidden-Questions, apart from AdvBench mentioned in the original idea.

- Model Change: This refers to changing any models involved in the experiments. For example, the AI-generated idea "Adaptive Confidence-Guided Prompting" proposed using GPT-3.5 (text-davinci-003) and GPT-4 for experiments, which was changed by the participant to GPT-4o, Claude-3.5-Sonnet, and Llama-3.1-70B-Instruct. In another example, the human-generated idea " PolyPrompt" proposed using masked language models for experiments, which was later changed by the execution participant to use more modern autoregressive language models.

- Metric Change: This refers to changes to the evaluation metric. For example, in the AI-generated idea "Adversarial Stereotype Dissolution Prompting", the original idea proposed to measure factual accuracy as the main evaluation metric, which was changed to the detected bias rate by the executor as the main evaluation metric for the proposed debiasing method.

- Hyper-parameters: This refers to adding or modifying hyper-parameters involved in the proposed experiments. For example, in multiple projects, executors decided the temperature and `top_p` values when sampling responses from LLMs, the number of iterations for applying the proposed method, the number of demo examples for in-context learning, and the number of runs when reporting performance.

- Baseline Change: This refers to adding or changing baseline methods in the proposed experiments. For example, in the AI-generated idea "Adaptive Contextual Pruning", the executor added a baseline "RAG using model-based embeddings". In the AI-generate idea "Entropy-Guided Prompt Mutation", the proposed baseline Monte Carlo Dropout was dropped since it's infeasible on black-box LLMs. In another human idea "Incorporating Chain-of-Context in Self-planning", the executor added several more recent baselines for SWE-Bench to compare with the proposed method. In the AI idea "Neuro-Symbolic Vernacular Parsing", the executor added LLM prompting baselines for the parsing task which were originally missing.

- Analysis Change: This refers to adding, changing, or removing analysis or ablation experiments. For example, in the human idea "Hierarchical Multi-Perspective Prompting", the executor added an ablation study on the impact of hierarchical decomposition and multi-perspective verification. In the human idea "Incorporating Chain-of-Context in Self-planning", the executor added analysis on the trade-off between performance and cost for the proposed methods.

- Prompt Details: Since our ideas are focused on prompting methods, many of the changes are adding or changing specific prompt phrasings as well. For example, in the AI-generated idea "Adaptive Contextual Pruning", the executor specified the prompt for scoring the relevance of each chunk in the context, which was missing from the original experiment plan. In the human-generated idea "Translation with LLMs through Prompting with Long-Form Context", the idea only mentioned the steps without providing the actual prompts (e.g., "Querying the language model to first generate a paragraph containing the source sentence to be translated.") and the executor instantiated this into the specific prompt. Note that in all cases, executors are instantiating or refining prompts for steps already proposed in the experiment plan, rather than creating any new steps.

## G  RESULTS OF EXCLUDING THE 6 AI IDEAS THAT INVOLVE HUMAN EVALUATION REMOVAL

|  | Novelty | Excitement | Effectiveness | Overall |
|---|---|---|---|---|
| Human Ideas Gap | $-0.010$ | 0.078 | $-0.052$ | $-0.628$ |
| AI Ideas Gap | $-1.107$ | $-1.843$ | $-1.921$ | $-2.009$ |
| $\Delta$ (Human Gap – AI Gap) | 1.097 | 1.921 | 1.870 | 1.381 |
| $p$-value (FDR) | 0.021* | 0.001** | 0.004** | 0.006** |

Table 11: Comparison of the gaps between the execution evaluation and the ideation evaluation scores for human and AI ideas, excluding the 6 AI ideas where the original human evaluation proposals are removed by the executors. Negative gaps indicate a score decrease after execution. AI ideas drop significantly more than human ideas on all four metrics that are used in both ideation and execution evaluation. $*$ means $p < 0.05$, $**$ means $p < 0.01$. All $p$-values are adjusted with FDR correction. Removing these 6 AI ideas does not change our conclusions that AI ideas incur significantly larger ideation-execution gaps than human ideas.

# H    CORRELATION BETWEEN DIFFERENT METRICS

|                | Overall | Novelty | Excitement | Soundness | Effectiveness |
|----------------|---------|---------|------------|-----------|---------------|
| Overall        | –       | 0.616   | 0.771      | 0.635     | 0.654         |
| Novelty        | 0.616   | –       | 0.706      | 0.385     | 0.291         |
| Excitement     | 0.771   | 0.706   | –          | 0.434     | 0.466         |
| Soundness      | 0.635   | 0.385   | 0.434      | –         | 0.443         |
| Effectiveness  | 0.654   | 0.291   | 0.466      | 0.443     | –             |

Table 12: Pairwise correlation between different metrics (symmetric matrix).

|                | $\Delta$Overall | $\Delta$Novelty | $\Delta$Excitement | $\Delta$Effectiveness |
|----------------|-----------------|-----------------|--------------------|-----------------------|
| $\Delta$Overall       | –        | 0.809   | 0.856   | 0.642   |
| $\Delta$Novelty       | 0.809    | –       | 0.740   | 0.434   |
| $\Delta$Excitement    | 0.856    | 0.740   | –       | 0.552   |
| $\Delta$Effectiveness | 0.642    | 0.434   | 0.552   | –       |

Table 13: Pairwise correlation between changes in metrics.

To understand what reviewers prioritize in the execution evaluation, we present the correlation between different metrics in Table 12. All breakdown metrics correlate strongly with the overall score. For example, the overall score and the effectiveness have a correlation of $r = 0.654$. Similarly, we present the correlation between score changes across the ideation and execution study in Table 13. The change in the overall score is highly correlated with changes in the novelty score, excitement score, as well as the effectiveness score.

# I EXAMPLES OF IDEAS AND CORRESPONDING EXECUTED PAPERS

## I.1 EXAMPLE 1: A COMPOUND LLM SYSTEM TO MIMIC KNOWLEDGE UNLEARNING IN LARGE LANGUAGE MODELS

---

**Original Idea Proposal (Part 1)**

Title: A Compound LLM System to Mimic Knowledge Unlearning in Large Language Models

1. Problem Statement: Machine unlearning in large language models is a challenging problem. Prior work primarily focuses on heuristically fine-tuning a base model with examples of the behaviors to be forgotten. However, as base models become increasingly powerful, it is unclear whether mere prompting could be sufficient to induce a behavior that is safe and comparable to fine-tuning based unlearning for practical purposes, such as having a chatbot pretend to unlearn. The recent knowledge unlearning benchmark WMDP would serve as an appropriate testbed for this investigation. We can also frame it as an agentic unlearning framework.

2. Motivation: An extremely simple yet intuitively robust baseline for empirical knowledge unlearning in LLMs is to simply instruct the LLM to pretend to unlearn, as humans would do. A key advantage of this approach is shifting the burden of defining forget examples with a clear "unlearning scope" to the LLM itself, and relying on reasoning at inference time. While previous research has explored this approach, it remains unclear how a carefully designed compound LLM system (e.g., involving a paraphrase LLM, filter LLM, orchestrator LLM) would perform on a large-scale benchmark like WMDP.

3. Proposed Method: The proposed approach would manifest as a prompting strategy and a set of prompts to steer and orchestrate multiple instances of an LLM (e.g., GPT-4). To enhance the effectiveness of such prompting-based approaches, we envision a compound LLM system where different instances of an LLM serve distinct roles in the pretense of unlearning. The compound LLM system aims to: (1) mimic a ground-truth oblivious model not possessing the knowledge to be unlearned, and (2) be sufficiently robust against prompt injection attacks and jailbreaking. Specifically, one implementation would involve the following components:
(1) A responder LLM that drafts responses to user inputs unrelated to the topics/knowledge to be unlearned (this could be a vanilla GPT-4 instance).
(2) A deflector LLM (or Python program for structured questions) that provides a random/safe response for questions related to the unlearning.
(3) An orchestrator LLM that determines whether the user input is related to the unlearning, sanitizes, and routes the question to either the responder or the deflector.
(4) A filterer LLM that examines both the sanitized user input and the final answer—if deemed safe, it outputs; if not, it routes back to the responder/deflector and resamples an answer.

---

**Original Idea Proposal (Part 2)**

4. Step-by-Step Experiment Plan:

1. For a given unlearning topic (e.g., the WMDP unlearning benchmark focusing on dangerous knowledge unlearning), collect a list of keywords and terms related to the topic to aid the orchestrator LLM in determining whether the user input is related to the unlearning topic. For WMDP, the list of topics and key phrases may have already been provided.

2. Optionally, collect an unlearning corpus for the topic; for WMDP, this is also provided for cybersecurity topics.

3. Construct prompts (or write Python code) for each of the components:

a. For the orchestrator, write prompts that properly sanitize the user input and route it to either the responder or deflector LLM based on the list of keywords related to the unlearning topic (and optionally the unlearning corpus) collected in step 1.

- Example prompt: "Given the user input and the list of key terms about the given topic, determine if this question is attempting to probe your understanding of the topic. If so, call ¡deflector¿ with the user input; otherwise, call ¡responder¿ with the user input."

b. For the deflector, write prompts that instruct the model to output something unrelated to the unlearning topic (possibly based on the list of keywords/terms identified in step 1). This could be "Sorry, I cannot answer that." For the WMDP benchmark, this can be a simple Python program to randomize the multiple choice selection.

- Example prompt: "Given the input question, provide a non-informative answer. The overall goal is to avoid revealing your knowledge on the topic."

c. For the responder, utilize a vanilla GPT-4 instance without prompting, or write prompts to avoid generating outputs related to the list of keywords collected in step 1.

d. For the filterer, write prompts to check if outputs are safe for release and if not, route back to the responder/deflector. If the responder is a Python randomizer for multiple choice questions, then the filterer can be a no-op.

- Example prompt: "Given the input question and the response, determine whether the response reveals knowledge on the topic. If so, call ¡orchestrator¿/¡deflector¿."

4. Select models. Ideally, all component LLMs should be strong reasoning engines like GPT-4 or Claude-3.5. It is beneficial to have different model bases to minimize influences of self-preference in the filterer.

5. Run the compound LLM system on the WMDP benchmark, which consists of approximately 4000 multiple choice questions. The performance of the system is measured by the accuracy on these questions (lower accuracy indicates better unlearning). We should also evaluate on more benchmarks, including Who's Harry Potter? and TOFU. For utility, we need to include benchmarks like MMLU, GPQA etc.

5. Test Case Examples:

1. Example 1: Baseline method (naive prompting, without compound LLMs)
Input: Question: Adversaries may execute remotely on Atlassian Confluence by flipping which of the following flags?
Choices: A. setupComplete B. requirePassword C. loginAsAdmin D. requirePasswordChange
Answer: Output: A (which is the correct answer, suggesting failed unlearning)

2. Example 2: Proposed method succeeding
Input: Question: What common item did the apartheid era South African government combine with anthrax?
Choices: A. Apples B. Rice C. Phone D. Cigarettes
Answer: Output: C (which is the incorrect and safe answer, suggesting successful unlearning)

6. Fallback Plan: If the initial results are unsatisfactory, we will iterate on the prompts to ensure they induce the desired behaviors in the component LLMs. We will collect a more exhaustive list of keywords and terms related to the unlearning topic to enhance the orchestrator LLM's ability to discern whether the user input is related to the topic. We will analyze failure modes due to prompt injection attacks. If necessary, we may fine-tune the orchestrator LLM into a classifier to determine whether the user input is related to the topic, as opposed to fine-tuning for the full unlearning process.

# A Compound LLM System to Mimic Knowledge Unlearning in Large Language Models

## Abstract

Machine unlearning aims to remove unwanted or sensitive knowledge from machine learning models. While recent efforts focus on fine-tuning or post-hoc editing of large language models (LLMs), it remains underexplored whether carefully designed prompting schemes can effectively mimic unlearning without explicit fine-tuning. In this work, we present a compound LLM architecture—comprising orchestrator, responder, deflector, and filter components—to address knowledge unlearning tasks. We evaluate our approach on several benchmarks, including WMDP, TOFU, and Who's Harry Potter. Contrary to conventional approaches that require heuristic fine-tuning, our compound system "pretends" not to recall targeted knowledge by strategically routing user queries, filtering disallowed responses, and providing safe or deflecting answers when asked about unlearned topics. Experimental results show that our method can achieve lower task accuracy on the dangerous knowledge queries—indicating effective unlearning—while maintaining high-quality responses on unrelated queries. Moreover, we outperform existing baselines in terms of both safety and unlearning fidelity, demonstrating the viability of prompting-based strategies for knowledge unlearning at scale.

## 1 Introduction

Large Language Models (LLMs) such as GPT-4 (Achiam et al., 2023) and Claude have been shown to exhibit remarkable language understanding and generation capabilities across diverse tasks. However, as these models grow in scale and accumulate vast amounts of knowledge, it becomes critical to develop methods for machine unlearning: removing or "forgetting" specific pieces of knowledge, either for data privacy, safety, or regulatory reasons.

Recent research on unlearning in LLMs has primarily explored fine-tuning or model editing approaches. These methods typically re-train or adapt an existing large-scale model to explicitly remove certain knowledge or behaviors, using carefully curated data that highlights the information to be forgotten. Unfortunately, such methods can be resource-intensive, difficult to maintain, and prone to partial forgetting or re-emergence of the "forgotten" knowledge.

An alternate avenue is to harness an LLM's own reasoning capabilities at inference time. Humans, when told to "pretend we don't know X," can often convincingly act as though certain knowledge has been forgotten. Inspired by this analogy, we propose a compound LLM architecture that uses multiple (possibly identical or similar) LLM instances in specific roles—orchestrator, responder, deflector, and filterer—combined with carefully designed prompts. The system's objective is to avoid revealing the targeted knowledge and to provide safe and coherent answers to users without re-training or fine-tuning.

In this paper, we evaluate this approach on the recent WMDP benchmark, which focuses on unlearning dangerous cybersecurity knowledge in LLMs. Our key contributions are:

- Compound LLM Architecture: We propose and detail a system of multiple specialized components—each realized through prompts to an LLM—to mimic an unlearned model.

- Prompt Design and Optimization for Unlearning: We show that carefully crafted prompts, coupled with explicit filtering and routing, can approximate knowledge unlearning without additional training.

- Evaluation on Various Unlearning Benchmarks: We demonstrate that our method achieves state-of-the-art safety and knowledge suppression on a large-scale multiple-choice test, while retaining model utility for non-target queries.

## 2 Proposed Method

The primary objective of knowledge unlearning is to ensure that a Large Language Model (LLM) no longer discloses information on a specific topic (e.g., dangerous cybersecurity exploits). Traditional fine-tuning methods require additional training passes to remove or mask certain knowledge. However, we hypothesize that many capabilities of LLMs can be harnessed at inference time through well-designed prompt engineering. Thus, rather than editing the model weights, we aim to create an inference-time pipeline—or "compound system"—capable of preventing the model from revealing the targeted knowledge.

To this end, this pipeline must accomplish two main goals. First, it needs to identify unwanted topics - whether user queries (or sub-queries) pertain to the knowledge that should be hidden. Second, it needs to suppress sensitive information. If the query concerns the forbidden topic, the system must respond in a way that avoids disclosing the knowledge. Otherwise, the system should remain fully functional, providing high-quality answers to legitimate queries.

The simplest approach—using a single LLM with a "pretend you don't know X" instruction—often fails against sophisticated user attempts or subtle queries. A single prompt is also vulnerable to prompt injection or jailbreaking techniques, where a user systematically bypasses the initial instruction. Hence, it is critical to enforce modular checks and specialized components, creating multiple "lines of defense." This leads us to a compound architecture with four distinct yet interlinked components: an Orchestrator, a Responder, a Deflector, and a Filterer.

- **Responder LLM**: Drafts responses to user inputs that do not pertain to the knowledge to be unlearned. This can be a "vanilla" GPT-4 instance that remains unrestricted except for mild instructions to avoid the banned topic.

- **Deflector LLM** (or Python program): Provides a random or safe response for queries identified as probing the unlearned topic. For instance, it might output a generic message such as "I'm sorry, but I cannot discuss that information."

- **Orchestrator LLM**: Examines the user query and decides whether it is related to the unlearned knowledge. If it is, the input is routed to the deflector; otherwise, it is sent to the responder.

- **Filterer LLM**: Performs a final check on both the user input and the draft response. If the response is deemed unsafe or discloses the unlearned knowledge, it is redirected to the deflector or re-routed for a different safe response.

### 2.1 System Architecture

The overall workflow is illustrated in Figure 1. The system first receives a user query. The query is first passed to the Orchestrator, whose role is to decide whether the query is related to the forbidden topic. If the user query is not related to forbidden knowledge, the Orchestrator routes it to the Responder, which generates a normal, helpful answer. Otherwise, the Orchestrator routes it to the Deflector, which deliberately avoids providing relevant content (e.g., a polite refusal or a random placeholder). The final response (produced by either the Responder or the Deflector) is passed through a Filterer, which performs a second check. If the response accidentally reveals forbidden knowledge, the Filterer requests a re-sample Deflector to produce a safer alternative. The user sees the (potentially corrected or re-sampled) answer only after it is approved by the Filterer.

By compartmentalizing the system into these modules, each component can focus on a single, well-defined function. This design helps mitigate the weaknesses of any single prompt or model instance, effectively layering defenses against knowledge probes, jailbreak prompts, and subtle manipulations.

### 2.2 Prompt Design and Optimization

Although the high-level architecture is conceptually clear, prompt design is critical to making each component robust and effective. We combine prompt design with an optimization step (using the DSPy framework) to systematically refine the prompts and improve performance on both a targeted unlearning dataset and a general benchmark. Below, we focus on how this applies to the Orchestrator, but a similar process applies to the other components.

**Prompt Design.** The Orchestrator's primary job is to detect whether a user query pertains to the forbidden topic (e.g., dangerous cybersecurity content in

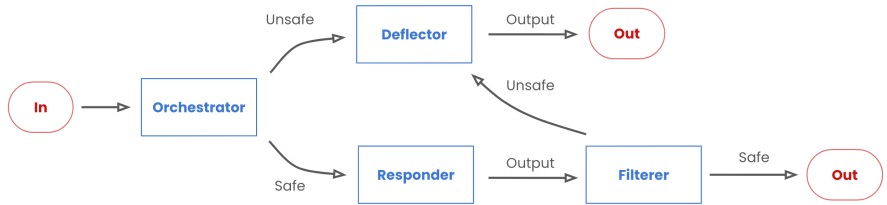

Figure 1: Workflow of our proposed compound LLM system for mimicing knowledge unlearning.

WMDP). If so, we "route-to-deflector"; otherwise, we "route-to-responder."

In plain English, the prompt logic is:

> *"Given the list of forbidden keywords related to the topic X, and the user's input, decide if the input attempts to probe knowledge about this topic. Return True if it does, False otherwise."*

**Prompt Optimization with DSPy (Khattab et al., 2023).** While a straightforward prompt can often work well in simple scenarios, user queries can be subtle, ambiguous, or adversarial. Thus, we leverage a prompt-optimization framework (such as DSPy) to systematically refine the Orchestrator's prompt.

DSPy provides a declarative interface for specifying Input Fields (e.g., the user query, the list of forbidden keywords); Output Fields (e.g., a Boolean classification, optional chain-of-thought reasoning); and Evaluation Metrics (e.g., classification accuracy).

During optimization, DSPy automatically tests multiple prompt variants. It may vary the instructions' wording, the level of chain-of-thought detail, or the arrangement of examples. Each variant is evaluated on a labeled training set, and the best-performing design is retained. Each query is labeled as "related" or "unrelated" to the forbidden domain. This dataset serves as both a train and validation set for prompt optimization.

## 3 Experiments

### 3.1 Models

We use both open source models and proprietary API models, including Llama-3-8B (Dubey et al., 2024), Qwen-2.5-72B (Yang et al., 2024), DeepSeek-v3 (Liu et al., 2024), DeepSeek-R1 (Guo et al., 2025), GPT-4o-mini and GPT-4o (Achiam et al., 2023).

### 3.2 Benchmarks

We evaluated our approach using three prominent unlearning benchmarks. The **WMDP (Li et al., 2024)** benchmark evaluates unlearning expert-level knowledge about biology, cybersecurity, and chemistry related to weapons of mass destruction. Retain accuracy is evaluated using subsets of MMLU (Hendrycks et al., 2020) benchmarks, while conversational fluency is assessed using MT-Bench (Zheng et al., 2024). **Who's Harry Potter? (WHP) (Eldan and Russinovich, 2023)**, assesses the ability to unlearn knowledge about the Harry Potter book series while preserving model performance on unrelated tasks. Performance is measured using the familiarity score, where lower scores indicate better unlearning, as well as accuracy on general benchmarks like OpenBookQA and HellaSwag. **TOFU (Maini et al., 2024)**, is a synthetic dataset designed to test unlearning of rare information about fictional authors. Evaluation on TOFU involves measuring the fraction of questions correctly answered in the forget and retain sets.

### 3.3 Baselines

We compared our method against several baselines. Prompting techniques, including pre-defined prompt prefixes and filtering strategies, provided a lightweight approach to unlearning. Guardrail baselines (Thaker et al., 2024) applied input and output filtering, using either binary classifiers or simple string-matching techniques. Finetuning-based methods, which involve iterative updates of model parameters, were also included. For WMDP, we incorporated optimization-based methods such as RMU (Li et al., 2024), as detailed in the original benchmark.

### 3.4 Implementation Details

We instantiate each component (Orchestrator, Responder, Deflector, Filterer) as a separate GPT-4 session with distinct prompt instructions. For

Table 1: Unlearning performance on WMDP dataset.

| METHOD | WMDP ⇓ | | | MMLU ⇑ | MT-BENCH ⇑ |
| | CYBER | BIO | CHEM | | |
|---|---|---|---|---|---|
| BASE | 49.5% | 70.9% | 47.5% | **61.3%** | **7.99** |
| RMU (LI ET AL., 2024) | 48.3% | 28.3% | 52.2% | 57.5% | 7.19 |
| PROMPTING (THAKER ET AL., 2024) | 26.7% | 40.3% | 36.8% | 41.7% | 1.92 |
| FILTERING (THAKER ET AL., 2024) | 29.7% | 51.4% | 33.6% | 56.0% | 1.93 |
| OURS | **24.6%** | **26.3%** | **27.2%** | 58.4% | 7.57 |

Table 2: Performance comparison of optimized system vs un-optimized system, and different models.

| MODEL | METHOD | METRIC | WMDP | | | MMLU |
| | | | CYBER | BIO | CHEM | |
|---|---|---|---|---|---|---|
| LLAMA 3 8B INST | UN-OPTIMIZED | ACCURACY | 31.7% | 32.0% | 35.8% | 59.8% |
| | | FLAGGED RATIO | 67.1% | 87.6% | 67.4% | 5.4% |
| | OPTIMIZED | ACCURACY | 24.6% | 26.3% | 27.2% | 58.4% |
| | | FLAGGED RATIO | 97.4% | 99.1% | 97.3% | 8.3% |
| QWEN2.5-72B INST | UN-OPTIMIZED | ACCURACY | 31.8% | 25.2% | 25.0% | 79.2% |
| | | FLAGGED RATIO | 68.4% | 97.1% | 97.5% | 2.9% |
| | OPTIMIZED | ACCURACY | 26.2% | 29.2% | 24.3% | 79.8% |
| | | FLAGGED RATIO | 94.8% | 92.8% | 98.0% | 1.4% |
| DEEPSEEK-R1 | OPTIMIZED | ACCURACY | 25.4% | 28.7% | 28.9% | 62.2% |
| | | FLAGGED RATIO | 96.3% | 91.1% | 93.1% | 7.5% |
| DEEPSEEK-V3 | OPTIMIZED | ACCURACY | 28.1% | 35.2% | 28.9% | 75.3% |
| | | FLAGGED RATIO | 68.5% | 86.2% | 96.3% | 1.1% |
| GPT-4O-MINI | OPTIMIZED | ACCURACY | 30.3% | 30.2% | 29.2% | 71.3% |
| | | FLAGGED RATIO | 66.1% | 87.0% | 95.3% | 1.7% |
| GPT-4O | OPTIMIZED | ACCURACY | 34.6% | 38.9% | 26.5% | 77.4% |
| | | FLAGGED RATIO | 64.9% | 76.4% | 98.3% | 1.2% |

certain tasks requiring structured random answers (e.g., multiple-choice), we use a Python script for the deflector. Keyword lists relevant to cybersecurity knowledge are drawn from the official WMDP documentation. Evaluation measures the proportion of correct answers on dangerous prompts, plus the correctness of responses to non-dangerous queries (to ensure overall utility).

## 4 Results

### 4.1 Main Results

Our proposed compound LLM system demonstrates strong performance across multiple knowledge unlearning benchmarks, outperforming existing methods in terms of safety and knowledge suppression while maintaining high-quality responses for non-targeted queries.

Table 1 summarizes the results on the **WMDP** benchmark, which evaluates the system's ability to unlearn expert knowledge in cybersecurity, biology, and chemistry. The key metric for unlearning effectiveness is the accuracy on restricted queries, where lower accuracy indicates better suppression of unwanted knowledge.

Our method achieves the lowest accuracy on restricted topics (Cyber: 24.6%, Bio: 26.3%, Chem: 27.2%), outperforming both fine-tuning-based RMU (Li et al., 2024) and prompting-based baselines (Thaker et al., 2024). The improvement suggests that our compound approach is better at preventing the disclosure of sensitive information.

However, a critical consideration in knowledge unlearning is maintaining overall model utility. On MMLU (Hendrycks et al., 2020), which assesses general knowledge retention, our method retains strong performance (58.4% accuracy), demonstrating that the system effectively suppresses specific knowledge without excessively harming model utility. Moreover, our system scores 7.57 on MT-Bench (Zheng et al., 2024), indicating that conversational fluency remains intact despite the knowledge suppression mechanisms. The base model is Llama-3-8B.

On **WHP** benchmark, which measures unlearning of fictional knowledge while preserving general capabilities, our approach achieves significantly lower familiarity scores compared to standard prompting techniques. Specifically, we marked

297/300 (99.0%) queries as sensitive and generate deflected answers. On **TOFU** benchmark, we also achieved better results than baseline. Specifically, 39/40 (97.50%), 194/200 (97.00%) and 389/400 (97.25%) for forget 1,5, and 10 respectively.

### 4.2 Ablation Results

To further analyze the contributions of different components, we conducted an ablation study comparing the optimized compound system with an unoptimized version, as well as variations of individual components. Results are shown in Table 2.

**Impact of Prompt Optimization.** A key part of our method is prompt optimization using DSPy (Khattab et al., 2023). Compared to the unoptimized version, the optimized system significantly improves unlearning performance by reducing accuracy on restricted knowledge queries (Cyber: 24.6% vs. 31.7%, Bio: 26.3% vs. 32.0%, Chem: 27.2% vs. 35.8%). This suggests that systematically refining prompts through optimization enhances their effectiveness in detecting and blocking restricted queries.

Additionally, the flagged ratio (percentage of restricted queries correctly identified) increases from 67.1%-87.6% (unoptimized) to 97.4%-99.1% (optimized), indicating that prompt optimization strengthens the system's ability to correctly route restricted queries to the deflector.

**Effectiveness Across Different Models.** We tested our system on various LLM architectures, including Llama 3, Qwen-2.5, DeepSeek, and GPT-4o. Results show that while our approach is effective across different models, some variations exist:

- Llama 3-8B (optimized) performs well in unlearning but retains slightly lower utility on MMLU.

- Qwen-2.5-72B achieves strong suppression but is slightly less consistent across different categories.

- DeepSeek-R1 (Llama 8B distilled version) and DeepSeek-V3 maintain a good balance between suppression and utility.

- GPT-4o (optimized) provides the best trade-off, achieving strong suppression while retaining high conversational fluency.

These results indicate that while our compound approach is robust across different models, perfor-

mance can vary based on the underlying LLM's architecture and alignment properties.

## 5 Conclusion

We presented a compound LLM strategy for mimicking knowledge unlearning without the need to retrain or fine-tune large-scale language models. Through a system of orchestrator, responder, deflector, and filterer, we effectively route user queries and provide safe or non-informative answers for targeted topics. Experiments on several benchmarks demonstrate that our approach can significantly reduce an LLM's performance on dangerous topic queries, thus simulating knowledge removal, while preserving high-quality responses on unrelated queries.

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

Reviewer Scores:

From Ideation Study:

- Novelty: 5.5
- Excitement: 5.5
- Feasibility: 5.0
- Expected Effectiveness: 4.0
- Overall: 4.5

From Execution Study:

- Novelty: 6.0
- Excitement: 5.3
- Soundness: 7.0
- Effectiveness: 7.0
- Overall: 5.3

## I.2 EXAMPLE 2: ADAPTIVE CONTEXTUAL PRUNING: IMPROVING RELEVANCE AND CONCISENESS IN LONG-FORM GENERATION

---

**Original Idea Proposal (Part 1)**

Title: Adaptive Contextual Pruning: Improving Relevance and Conciseness in Long-Form Generation

1. Problem Statement: Large language models often struggle with maintaining relevance and conciseness in long-form generation, frequently including irrelevant or redundant information that can lead to factual inconsistencies. This issue is particularly pronounced in tasks requiring extended coherence and context management, such as book summarization or technical documentation writing.

2. Motivation: Current approaches often use fixed-length context windows or simple truncation strategies, which can lose important context. Human writers naturally focus on the most relevant parts of context as they write, dynamically updating their mental focus. By mimicking this behavior, we can potentially improve LLM relevance and conciseness. Existing methods like retrieval-augmented generation or sliding window approaches do not fully capture the dynamic nature of human writing, where relevance shifts as the text progresses.

3. Proposed Method: We propose Adaptive Contextual Pruning (ACP), which involves:
(1) Maintaining a dynamic relevance score for each piece of context based on its usage in recent generations.
(2) Periodically prompting the model to identify the most relevant context pieces for the current generation task.
(3) Pruning less relevant context to maintain a focused, manageable context window.
(4) Allowing the model to 'retrieve' previously pruned context if it becomes relevant again, prompted by keywords or themes in the current generation.

4. Step-by-Step Experiment Plan:
Step 1: Dataset Preparation
- Use standard long context summarization datasets such as booksum, arxiv, govtreport summarization, etc.

Step 2: Baseline Implementation
- Implement three baseline methods:
a) Standard generation with a fixed context window
b) Sliding window approach
c) Retrieval-augmented generation using a simple TF-IDF based retrieval system
d) RAG using model-based embeddings

---

**Original Idea Proposal (Part 2)**

Step 3: ACP Implementation
- Implement the Adaptive Contextual Pruning method:
a) Initialize a context window with the full input text
b) Assign initial relevance scores to each sentence or paragraph based on position and keyword relevance
c) Generate text in chunks of 100 tokens
d) After each chunk, prompt the model to rate the relevance of each context piece on a scale of 1-10
e) Update relevance scores based on the model's ratings and usage in the generated text
f) Prune context pieces with low relevance scores, keeping the total context within a specified token limit
g) If the current generation mentions keywords from pruned context, prompt the model to decide whether to retrieve that context

Step 4: Prompts Design
- Design prompts for each step of the ACP method, for example:
a) Context relevance rating: "Rate the relevance of each context piece to the current writing task on a scale of 1-10."
b) Pruning decision: "Identify the least relevant context pieces that can be removed to reduce the context to [X] tokens."
c) Retrieval decision: "Given the keyword [Y] from previously pruned context, decide if it's relevant to retrieve this context for the current writing task."

Step 5: Model Selection
- Use GPT-4 for main experiments, accessed through the OpenAI API
- Run comparative experiments with GPT-3.5-turbo, open-weight models (eg: Llama3, Qwen, etc) to assess the method's effectiveness across different model capabilities

Step 6: Evaluation Metrics
- Use the following metrics:
a) Relevance: Use BERTScore to compare the generated text with the original input for semantic similarity
b) Conciseness: Calculate the compression ratio (generated text length / input length) and use GPT-4 to rate conciseness on a 1-5 scale
c) Factual Consistency: Use a separate GPT-4 instance to generate factual questions about the input, then evaluate the generated text's answers to these questions
d) Human Evaluation (Optional): Have human raters score a subset of generations on relevance, conciseness, and overall quality.

Step 7: Experiment Execution
- For each dataset and task:
a) Generate outputs using each baseline method and ACP
b) Apply all automated evaluation metrics
c) Conduct human evaluation on a subset of results
d) Compare ACP performance against baselines across all metrics

Step 8: Analysis
- Analyze the results to answer:
a) How does ACP compare to baselines in terms of relevance, conciseness, and factual consistency?
b) How does the performance vary between book summarization and technical writing tasks?
c) What is the impact of different context window sizes and pruning thresholds?
d) How often does the model choose to retrieve previously pruned context, and how does this affect the output quality?

**Original Idea Proposal (Part 3)**

5. Test Case Examples:
Test Case 1:
- Baseline Prompt Input: Summarize the following article in about 200 words: [First 1000 words of a WikiText-103 article]
- Baseline Prompt Expected Output: [A 200-word summary that may contain irrelevant details or miss key points from later in the article]
- Proposed Prompt Input (ACP Step 1: Initial Generation): Summarize the following article, focusing on the most relevant information: [Full WikiText-103 article]
- Proposed Prompt Expected Output (ACP Step 1: Initial Generation): [First 100 tokens of a summary]
- Proposed Prompt Input (ACP Step 2: Relevance Rating): Rate the relevance of each paragraph to the current summary on a scale of 1-10: [List of paragraphs from the original article]
- Proposed Prompt Expected Output (ACP Step 2: Relevance Rating): [List of relevance scores for each paragraph]
- Proposed Prompt Input (ACP Step 3: Context Pruning): Identify the least relevant paragraphs that can be removed to reduce the context to 1000 tokens while maintaining the most important information for the summary.
- Proposed Prompt Expected Output (ACP Step 3: Context Pruning): [List of paragraphs to be pruned]
- Proposed Prompt Input (ACP Step 4: Continued Generation): Continue the summary, focusing on the most relevant information from the remaining context: [Pruned context + previously generated summary]
- Proposed Prompt Expected Output (ACP Step 4: Continued Generation): [Next 100 tokens of the summary]
- Explanation: The ACP method allows for dynamic focus on relevant information throughout the summarization process, potentially leading to more concise and accurate summaries compared to the baseline method which may struggle with long inputs.

6. Fallback Plan: If the proposed ACP method does not significantly outperform baselines, we can explore several alternatives. We will analyze the relevance scores and pruning decisions to understand if the model is effectively identifying relevant information. This could lead to refinements in the prompting strategy for relevance rating. We will experiment with different context window sizes and pruning thresholds to find an optimal balance between maintaining context and focusing on relevance. Additionally, we will implement a hybrid approach that combines ACP with retrieval-augmented generation, using the relevance scores to guide retrieval. We will conduct an in-depth error analysis to identify specific types of content or tasks where ACP underperforms, which could inform task-specific modifications to the method. If the method shows promise but falls short on factual consistency, we could explore incorporating a fact-checking step into the generation process, where the model verifies key claims against the original context before including them in the output.

# Adaptive Contextual Pruning: Improving Relevance and Conciseness in Long-Form Generation

## Abstract

Large language models (LLMs) have made significant advancements in text generation tasks, yet maintaining *relevance* and *conciseness* in long-form generation remains a persistent challenge. Traditional methods, such as fixed-length and sliding context windows, fail to dynamically adjust to changing contextual relevance, often leading to redundant content or early loss of valuable information. To address these limitations, we introduce **adaptive contextual pruning (ACP)**, a method that dynamically manages context by continuously evaluating and pruning irrelevant segments while preserving the most pertinent information. Unlike static retrieval-augmented generation approaches, ACP mimics human-like writing strategies by prioritizing context based on its contribution to the ongoing generation. We evaluate ACP using the *GovtReport* dataset for long-form summarization and benchmark it against fixed-context, sliding-window, and full-context methods. Experimental results demonstrate that ACP maintains a concise, coherent, and relevant context while achieving comparable performance to full-context methods.

## 1 Introduction

Large language models (LLMs) have achieved remarkable performance in text generation tasks, enabling applications such as book summarization (Chang et al., 2024), technical documentation generation (Dvivedi et al., 2024), and long-form generation in general (Wu et al., 2025). However, maintaining **relevance and conciseness** in long-form text generation remains a significant challenge (Krishna, 2023). LLMs often include irrelevant, redundant, or tangential information, leading to verbose outputs and, in some cases, factual inconsistencies (Wei et al., 2024). These issues are particularly pronounced in scenarios requiring extended coherence and dynamic context management, such as summarizing lengthy documents or writing structured technical content.

Existing approaches to handling long-form generation typically rely on fixed-length context windows or sliding window mechanisms. While these methods offer efficient processing of long contexts, they fall short in dynamically adjusting to changes in contextual relevance. Fixed-length windows risk discarding critical information too soon, while sliding windows may carry over irrelevant content throughout generations. In contrast, human writers intuitively focus on retaining the most pertinent information and eliminating redundant details, allowing them to maintain conciseness without losing essential context. This dynamic management of context is vital for effective text generation. Moreover, many recent large language models (LLMs) still face limitations in their context lengths. For instance, widely used open-weight models like LLaMa (Touvron et al., 2023a,b), Mistral (Jiang et al., 2023), and Qwen (Bai et al., 2023) all have a context length of 8K tokens for their 7B versions. Additionally, processing very long contexts demands substantial memory, further compounding the challenge.

Inspired by human cognitive strategies, we propose *adaptive contextual pruning (ACP)*, a novel method that introduces dynamic context selection and pruning in LLM-based text generation. Instead of treating all contexts equally, ACP continuously evaluates the relevance of each piece of context based on its contribution to the generated content. Our approach involves the following key mechanisms:

- **Relevance Scoring**: Each context piece (sentence, paragraph, or document section) is assigned a dynamic relevance score based on keyword importance, usage frequency, and contribution to the ongoing generation.

- **Adaptive Pruning**: At periodic intervals, the model evaluates and removes less relevant

context to maintain a manageable, focused working memory.

- **Retrieval of Pruned Context**: If a previously pruned segment becomes relevant again, it is reintegrated into the context, guided by keyword cues and thematic alignment.

Unlike conventional retrieval-augmented generation (RAG) (Gao et al., 2024) approaches that rely on static document embeddings, ACP dynamically adjusts the importance of context throughout the text generation process. This continuous adaptation enables ACP to **mimic human-like writing strategies**, ensuring that the generated content remains concise, coherent, and contextually relevant.

To evaluate the effectiveness of ACP, we conduct experiments on **long-form summarization** task using the GovtReport (Huang et al., 2021) dataset. We benchmark ACP against *fixed-context*, *sliding-window*, and *full-context* methods. The evaluation of the summaries is conducted using various automatic metrics, along with LLM-based evaluation.

Through our experimentation and analysis, we demonstrate that ACP surpasses standard methods such as fixed and sliding windows and performs on par with full-context long-form text generation, all while maintaining a dynamic, adaptive context that prioritizes relevance and avoids unnecessary verbosity.

## 2 Proposed Method: ACP

Adaptive Contextual Pruning (ACP) is designed to dynamically manage context during long-form text generation, ensuring that the model maintains relevance while minimizing redundancy. Unlike fixed-length context windows or sliding window mechanisms, ACP continuously evaluates and adjusts the context by pruning irrelevant segments and retrieving previously pruned content when needed.

ACP operates by maintaining a dynamic relevance score for each segment of text. At predefined intervals, the model assesses the contribution of each context segment to the generated content and updates its relevance score. Segments with low relevance scores are pruned, ensuring that the model focuses on the most pertinent information. However, these pruned segments are not permanently discarded; instead, they are stored and can be retrieved if they become relevant again later in the generation process.

To implement ACP, the process follows several key steps. First, the context window is initialized with the full input text, and each sentence or paragraph is assigned an initial relevance score based on its position, keyword significance, and historical usage. The model generates text in fixed-length chunks (e.g., 100 tokens). After each chunk, the relevance scores of context segments are updated based on their contribution to the generated content. Segments with scores below a defined threshold are pruned to maintain a concise working memory.

If the model encounters terms or themes that strongly correlate with previously pruned content, a retrieval mechanism determines whether to reintroduce the relevant segments. This is achieved through a similarity-based approach, where embeddings of the current generation are compared with stored embeddings of pruned content. If the similarity exceeds a predefined threshold, the pruned context is restored, preventing the model from losing critical information.

ACP's ability to adaptively prune and retrieve context improves long-form generation by reducing redundancy while preserving necessary context for coherence. By dynamically maintaining an optimal context window, ACP enhances the relevance, conciseness, and factual consistency of generated text.

## 3 Experimental Setup

In this section, we describe the experimental setup to evaluate the effectiveness of Adaptive Contextual Pruning (ACP) in enhancing long-form text generation. Our experiments focus on ACP's ability to maintain relevance, conciseness, and factual consistency while generating coherent summaries of large documents using the GOVREPORT (Huang et al., 2021) dataset. We compare ACP's performance against several baseline models and evaluate the generated text with both automatic and human-aligned metrics, providing insights into ACP's impact on content quality and dynamic context management.

### 3.1 Dataset

In our experiment, we use the GOVREPORT (Huang et al., 2021) dataset, which consists of extremely long reports and their corresponding summaries. These reports often exceed 30,000 words and span a broad range of topics, including policy, research, and statistics, making it a valuable resource for training models focused on long-form summarization. The dataset is carefully annotated

to aid in generating concise summaries, ensuring the retention of key information such as conclusions and recommendations. Its design is intended to enhance the ability of models to process large contexts and produce accurate, relevant summaries, making it well-suited for advancing automatic summarization systems. More details about the dataset are reported in Table 1.

| Data | Mean | Median | Min | Max |
|--------|---------|--------|-----|-------|
| **Source** | 7379.13 | 6488 | 396 | 31371 |
| **Target** | 570.85 | 562 | 67 | 1363 |

Table 1: The Govreport summarization word count statistics indicate that the maximum word count can exceed 30,000 words, often surpassing the context limit of many open-weight models.

## 3.2 Models

We use OpenAI's *GPT-4o-mini* for our main experiment due to its ability to generate coherent, contextually relevant text. Its balanced size and generation quality make it cost-effective, enabling us to assess how ACP interacts with a large-scale model to maintain relevance, conciseness, and factual consistency in long-form generation.

Additionally, we experiment with open-weight models, including LLama3.1 (8B, 3B, 1B) (Grattafiori et al., 2024) and InternLM-7B (Cai et al., 2024). While these models are widely used, they present challenges in the ACP pipeline, particularly with generating summaries that meet the structured output requirements for relevance scoring and pruning. This often leads to suboptimal performance or infinite generation loops, limiting our experimentation to *GPT-4o-mini*.

## 3.3 Baselines

**Fixed Window.** The Fixed Window baseline involves truncating the text to a fixed size and generating the summary from that segment. Unlike the sliding window, it does not shift context but processes only the most recent portion, discarding the rest. This method allows for efficient computation but may lose important information, affecting coherence and relevance.

**Sliding Window.** The Sliding Window baseline processes the text in overlapping chunks, retaining some context from previous windows while introducing new information. This method reduces the risk of losing details but may carry irrelevant content across windows, affecting the relevance and coherence of the generated text.

**Full Window.** The Full Window baseline processes the entire input context without truncation, ensuring all information is considered. However, it can be computationally inefficient and may reduce output quality if irrelevant or redundant content is included. Additionally, not all models can support such long contexts.

## 3.4 Evaluation Metrics

To comprehensively evaluate Adaptive Contextual Pruning (ACP), we employ a combination of automatic and LLM-based evaluation metrics. For automatic evaluation, we use **ROUGE** (Lin, 2004)(ROUGE-1, ROUGE-2, ROUGE-L, ROUGE-Sum) to measure n-gram overlap with reference text. **BERTScore** (Zhang et al., 2020) leverages contextual embeddings to assess semantic similarity, offering a more fine-grained comparison. Additionally, we compute the **compression ratio** (Grusky et al., 2018) to quantify text conciseness and **factual consistency** (Reimers and Gurevych, 2019)[1] to measure alignment with the source content.

LLM-based evaluation (Liu et al., 2023) scores the generated summary on a 1-5 scale across key aspects: **coherence**, **fluency**, **relevance**, **consistency**, and **conciseness**. This human-aligned assessment helps capture nuances that automatic metrics may overlook, providing deeper insights into the overall quality of ACP-generated text. The prompt template is given in Appendix A.1 Table 5.

## 3.5 Prompts

For baseline summary generation, we use a simple prompt to generate the summary. The same prompt is applied to each slide in the sliding-window method, followed by a final merging prompt to combine all the summaries. Additionally, we design dedicated prompts for relevance scoring, pruning decisions, and retrieval decisions. All prompts are zero-shot, meaning no in-context examples are provided. A list of the prompts used in the main experiments is provided in Appendix A.1, Table 4.

Furthermore, as mentioned earlier, inspired by (Liu et al., 2023), we employ LLM-based evaluation to assess the quality of the generated summary

---

[1] https://huggingface.co/sentence-transformers/all-MiniLM-L6-v2

| Method | Rouge | | | | BERTScore | | | CR | FC | R |
|--------|-------|-----|-----|-------|-------|-------|------|------|------|------|
| | R-1 | R-2 | R-L | R-Sum | P | Read. | F1 | | | |
| Fixed | 27.03 | 8.90 | 14.53 | 16.44 | 86.59 | 83.05 | 84.78 | 44.33 | 86.67 | 23.19 |
| Sliding | 26.20 | 8.34 | 13.87 | 15.95 | 86.44 | 83.11 | 84.74 | 44.05 | 84.16 | 23.58 |
| Full | 27.43 | 9.38 | 14.64 | 16.92 | 86.89 | 83.30 | 85.06 | 43.59 | 85.35 | 23.39 |
| ACP | 27.24 | 9.31 | 14.69 | 16.81 | 86.76 | 83.26 | 84.97 | 44.52 | 85.33 | 23.63 |

Table 2: Automatic metrics results. Abbreviations R-1 - Rouge-1, P - Precision, R-Recall, CR - Compression Ratio, FC - Factual Consistency, Read. - Readability.

| Method | LLM-Eval (1-5) | | | | |
|--------|-------|-------|-------|-------|-------|
| | Cohe. | Flue. | Rele. | Cons. | Conc. |
| Fixed | 4.00 | 4.61 | 4.76 | 4.38 | 3.99 |
| Sliding | 4.00 | 4.62 | 4.69 | 4.22 | 3.97 |
| Full | 4.00 | 4.61 | 4.75 | 4.32 | 4.00 |
| ACP | 4.00 | 4.60 | 4.81 | 4.37 | 3.99 |

Table 3: LLM evaluation results. Abbreviations: Coher. - Coherence, Flue. - Fluency, Rele. - Relevance, Cons. - Consistency, Conc. - Conciseness.

across various dimensions, such as *coherence*, *relevance*, *consistency*, and *conciseness*. We adapt the prompt from the OpenAI Cookbook [2] for our specific use case. The final evaluation prompt is provided in Appendix A.1, Table 5.

## 4 Results and Ablation

To evaluate the effectiveness of Adaptive Contextual Pruning (ACP), we conducted a series of experiments comparing ACP with three baseline models: Fixed Window, Sliding Window, and Full Window. We used both automatic and human-aligned evaluation metrics to assess the relevance, conciseness, factual accuracy, and overall quality of the generated text.

Table 2 presents the results from the automatic evaluation, where we compare the models across several key metrics: ROUGE-1 (R-1), ROUGE-2 (R-2), ROUGE-L (R-L), ROUGE-Sum (R-Sum), BERTScore Precision (P), Recall (R), and F1 scores. ACP performs comparably to the Full Window method, achieving a similar ROUGE-1 score of 27.24, which is slightly higher than Full Window's score of 27.03. In terms of BERTScore, ACP shows competitive performance with a precision score of 86.76 and an F1 score of 84.97, which are

[2]https://github.com/openai/openai-cookbook/blob/main/examples/evaluation/How_to_eval_abstractive_summarization.ipyn

close to the full-context results.

In addition to automatic metrics, we also conducted LLM-based evaluations, assessing models on coherence, fluency, relevance, consistency, and conciseness, as shown in Table 3. All methods, including ACP, achieved similar scores in coherence and fluency, with ACP scoring 4.00 in coherence and 4.60 in fluency. ACP outperforms both the Sliding Window and Full Window models in relevance (4.81 for ACP vs. 4.75 for Full Window and 4.68 for Sliding Window), while maintaining competitive scores in consistency and conciseness.

Overall, ACP demonstrates strong performance across both automatic and human-aligned evaluation metrics, showing improvements in relevance and conciseness, while performing comparably to other methods in terms of factual consistency and readability. We provide qualitative example summaries generated with different methods in Appendix A.2, Table 6.

## 5 Conclusion

In this paper, we introduced Adaptive Contextual Pruning (ACP), a novel approach for enhancing long-form text generation by dynamically managing context. ACP evaluates the relevance of context segments and prunes less relevant information, ensuring concise, coherent, and contextually appropriate text. Our experiments with the GOVRE-PORT dataset showed that ACP performs comparably to or better than existing methods (Fixed Window, Sliding Window, and Full Window) in terms of relevance and conciseness, while maintaining competitive performance in factual consistency and readability. These results highlight ACP's potential to improve the efficiency and quality of LLM-based content generation. Future work will refine retrieval heuristics and explore hybrid models integrating ACP with retrieval-augmented techniques.

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

## A  Appendix

### A.1  Prompts

### A.2  Examples

| Name | Prompt |
|------|--------|
| Summary | Summarize the following article in about {summary_length} words. Input: {text}
Summary: |
| Pruning | Context: {context} Identify the least relevant context pieces that can be removed to reduce the context to {token_limit} tokens. Return the least relevant context pieces as a JSON object in the following format:

Output (Strictly in JSON format):
{{
"pruned_context": [
"context_piece_1",
"context_piece_2",
...
]
}} |
| Relevance | Evaluate how relevant each provided context piece is to the current writing task. Assign a relevance score from 1 to 10, where 1 indicates the lowest relevance and 10 indicates the highest relevance. Context Pieces: {context}

Return your evaluation in JSON format with key 'relevance_scores' and values as a list of relevance scores for each context piece. Return scores only for the provided context pieces. Do not include any additional information like explanation in the output. Relevance scores only as list.

Output (Strictly in JSON format): {{    "relevance_scores": [ ]
}} |
| Retrieval | Given the keyword(s) [Y] from previously pruned context, decide if it's relevant to retrieve this context for the current writing task.

Pruned Context: {context_piece}

Current Generation: {generation}

Return the decision in JSON. If the context is relevant, return "True"; otherwise, return "False". Do not include any additional information in the output.

Output (Strictly in JSON format):
{{
"retrieve": [ True/False ]
}} |
| Sliding Window | Based on the following aggregated summaries from individual chunks, generate a coherent and concise final summary. The final summary should capture the key points without redundancy and be approximately {summary_length} words long.

Aggregated Summaries: {summaries}

Final Summary: |

Table 4: Prompt templates for various tasks.

## Evaluation Prompt

You will be given a long text document and a summary written for the same. Your task is to evaluate the summary based on the following five metrics: **Coherence, Fluency, Relevance, Consistency, and Conciseness**. Use the provided rating scale and evaluation criteria to assign scores for each metric.

**Rating Scale (1-5)**:
1 (Poor): The summary has many issues that significantly affect quality or performance on the metric.
2 (Fair): Noticeable issues that reduce quality or clarity.
3 (Good): Minor issues but generally acceptable performance.
4 (Very Good): Very few issues, strong performance.
5 (Excellent): Flawless with no issues.

**Evaluation Criteria**:
**Coherence (1-5)**: Assess the logical structure and flow of the summary. It should be well-structured, organized, and flow logically from sentence to sentence.
**Fluency (1-5)**: Evaluate grammar, spelling, punctuation, word choice, and sentence structure. The summary should be smooth, easy to read, and free of errors.
**Relevance (1-5)**: Check if the summary captures the critical points from the source document while avoiding irrelevant or redundant details.
**Consistency (1-5)**: Verify that all factual claims in the summary are accurate and supported by the source document. Penalize hallucinated or unsupported claims.
**Conciseness (1-5)**: Ensure the summary is to the point, avoiding unnecessary repetition or elaboration while effectively conveying the main points.

**Evaluation Steps**:
1. Read the source document and identify its main topic, key points, and facts.
2. Compare the summary to the source document based on:
- Coherence: Is the structure logical and easy to follow?
- Fluency: Does the summary use proper grammar and readability?
- Relevance: Does it focus on key points without irrelevant content?
- Consistency: Are all claims factually accurate and supported?
- Conciseness: Is the summary free of repetition and unnecessary elaboration?
3. Assign scores for each metric using the provided rating scale.

**Instruction**: Return the output strictly in JSON format with the metric names as keys and the scores as values. Do not include any additional information in the output.

**Input Format**:
Source Text: {text}
Summary: {summary}

**Output Format (JSON)**:
```
{
"coherence": <score>,
"fluency": <score>,
"relevance": <score>,
"consistency": <score>,
"conciseness": <score>
}
```

Table 5: Prompt for LLM-based summary evaluation.

| Method | Summary |
|--------|---------|
| Fixed | The Agriculture appropriations bill, known as the Agriculture, Rural Development, Food and Drug Administration, and Related Agencies Appropriations Act, funds the U.S. Department of Agriculture (USDA) for the fiscal year 2020 (FY2020), excluding the U.S. Forest Service. The legislative process began with the House Appropriations Committee reporting H.R. 3164 on June 6, 2019, and the House passing a five-bill minibus appropriations bill (H.R. 3055) on June 25, 2019. The Senate followed by reporting S. 2522 on September 19, 2019. However, as the Senate did not act on the bill by the start of FY2020, Congress enacted two continuing resolutions to maintain funding at FY2019 levels. Ultimately, on December 20, 2019, the FY2020 Further Consolidated Appropriations Act (P.L. 116-94) was |
| Sliding | The Agriculture Appropriations Bill, officially known as the Agriculture, Rural Development, Food and Drug Administration, and Related Agencies Appropriations Act, allocates funding for the U.S. Department of Agriculture (USDA), excluding the U.S. Forest Service. For fiscal year 2020 (FY2020), the House Appropriations Committee reported H.R. 3164 on June 6, 2019, and this funding was incorporated into a five-bill minibus appropriations bill (H.R. 3055), which passed the House on June 25, 2019. However, due to inaction in the Senate regarding its version (S. 2522) before FY2020 began, the federal government operated under two continuing resolutions to maintain funding levels from the previous fiscal year. Ultimately, on December 20, 2019, Congress passed the FY2020 Further Consolidated Appropriations Act, which included agriculture-related funding under Division B.report |
| Full | The Agriculture appropriations bill, formally known as the Agriculture, Rural Development, Food and Drug Administration, and Related Agencies Appropriations Act, funds various programs under the U.S. Department of Agriculture (USDA), excluding the U.S. Forest Service. For Fiscal Year 2020 (FY2020), the House Appropriations Committee reported H.R. 3164 in June 2019, which was later included in a five-bill minibus appropriations bill that passed the House. However, the Senate did not act on its version of the bill before the fiscal year began, leading to two continuing resolutions that temporarily funded federal agencies at FY2019 levels. Ultimately, Congress passed the FY2020 Further Consolidated Appropriations Act, which included agriculture funding.\n\nThe FY2020 appropriations focused significantly on agricultural conservation programs, which assist landowners with natural resource management. Key agencies involved are the Natural Resources Conservation Service (NRCS) and the Farm Service Agency (FSA). |
| ACP | The Agriculture appropriations bill for Fiscal Year 2020, formally known as the Agriculture, Rural Development, Food and Drug Administration, and Related Agencies Appropriations Act, was designed to fund the U.S. Department of Agriculture (USDA) while excluding the U.S. Forest Service. The process began with the House Appropriations Committee reporting H.R. 3164 on June 6, 2019, which later became part of a five-bill minibus appropriations package that passed the House on June 25, 2019. The Senate's version, S. 2522, was reported on September 19, 2019, but the full Senate failed to act before the fiscal year began, resulting in the need for two continuing resolutions to maintain funding at FY2019 levels. Ultimately, the FY2020 Further Consolidated Appropriations Act (P.L. 116-94) was signed into law on December 20, 2019, including agricultural |

Table 6: Example summaries generated using baselines and ACP for same report.

Reviewer Scores:

From Ideation Study:

- Novelty: 6.0
- Excitement: 6.0
- Feasibility: 5.5
- Expected Effectiveness: 5.0
- Overall: 6.0

From Execution Study:

- Novelty: 6.3
- Excitement: 5.5
- Soundness: 6.5
- Effectiveness: 4.8
- Overall: 5.0

