# OpenReview forum: "The Ideation-Execution Gap: Execution Outcomes of LLM-Generated versus Human Research Ideas"
_ICLR.cc/2026/Conference — ICLR 2026 Poster_

### Official Review · Reviewer_qsa4 · 2025-10-29

**Soundness:** 3
**Presentation:** 3
**Contribution:** 3
**Rating:** 6
**Confidence:** 4

**Summary:**

This paper presents the first large-scale execution study comparing LLM-generated and human-generated NLP research ideas. 43 expert researchers executed ideas (24 by AI, 19 by human) drawn from a prior ideation study. 58 expert reviewers then conducted double-blind reviews (181 in total) rating novelty, excitement, soundness, effectiveness, and overall quality.

Results reveal a clear ideation–execution gap: although LLM ideas scored higher during ideation, their scores dropped substantially after execution (≈1–2 points on a 1–10 scale), while human ideas remained stable. Post-execution, human ideas slightly outperformed AI ideas, reversing the earlier ranking. Analyses show that LLM ideas often suffered from unrealistic scope and weaker empirical design, whereas reviewers of executed work emphasized feasibility and soundness over speculative novelty.

**Strengths:**

- The work addresses whether LLMs can generate effective research ideas, not just novel ones. The study uses randomized assignment, double blinding, standardized instructions, and FDR-corrected statistical testing.

- The work combines quantitative results with qualitative review comment analysis, explaining why AI ideas underperform in execution.

- It establishes an “execution-aware” evaluation paradigm and commits to open-sourcing all data, code, and reviews.

**Weaknesses:**

- 43 projects across seven NLP topics limit generalization; authors acknowledge this.

- No contamination or recency discussion: Claude 3.5-Sonnet was used, but its training cutoff and overlap with prior research are not analyzed.

- Reviewer consistency is reported only in aggregate; topic-level calibration is missing.

- Overall technical contribution is not significant.

**Questions:**

- How is training-data cutoff handled—any risk of contamination?

- Would results differ with newer models (e.g., GPT-4o)?

- Could a predicted “feasibility” score reduce the gap?

- According to Table 3, AI idea executors are more experienced than those for Human. Would this affect the results?

---

> ### Author Response · Authors · 2025-11-29
> **Response to Reviewer qsa4 (Part 1)**
>
> 1. *How is training-data cutoff handled—any risk of contamination? / No contamination or recency discussion: Claude 3.5-Sonnet was used, but its training cutoff and overlap with prior research are not analyzed.*
>
> In the idea generation process, we had a final filtering step where we retrieved similar papers to the generated idea (via keyword search through the Semantic Scholar API) and asked an LLM judge to determine whether the idea is the same as any of them. We filtered out ideas that are judged as equivalent to any of the retrieved papers. That being said, our novelty filter is by no means perfect, and there are still potential cases where the model is rephrasing ideas from papers in the training data.
>
> That’s why we recruited expert reviewers to judge the novelty score of all executed ideas. If any LLM-generated ideas are from the training data, it should be reflected by a lower novelty score. As a rough estimate, out of the 181 free-text reviews on the novelty metric, 100 of them mentioned at least one prior paper to justify their novelty score (this is based on a simple keyword search through keywords like “arxiv.org” and “aclanthology.org” as a conservative estimate). Some examples of the actual reviews include:
> - “Examples of more robust and similarly idealized techniques are: https://arxiv.org/pdf/2410.08660 retrieves examples of pre-collected jailbreaks that were decomposed to teach LLMs to separate malicious content.”
> - “there're similar ideas in existing literature, e.g. this paper called Prompting Techniques for Reducing Social Bias in LLMs through System 1 and System 2 Cognitive Processes (https://arxiv.org/pdf/2404.17218).”
> - “the stepwise part has been explored in the literature too (e.g. Hiyati et al.,24; https://arxiv.org/abs/2311.09799)”
> - “The paper basically proposes IR-COT(https://arxiv.org/abs/2212.10509), which is an ACL 2023 paper, but for a different use case (proverb translation v.s. multi-hop QA).”
> - “I believe the core idea of this paper --- using a self-reflection process to detect and correct biases -- has been implemented in other works. For example, work from Liu et al. \"Self-Reflection Makes Large Language Models Safer, Less Biased, and Ideologically Neutral\" implement a self-reflection technique to mitigate biases in language models.”
> - “There's already a lot of work looking at the idea of breaking down complex concepts and questions into simpler ones and re-composing in prompting (e.g. this: https://aclanthology.org/2022.emnlp-main.81/).”
> - “The idea of combining iterative retrieval and self-planning for code generation has been explored in prior work (e.g., https://arxiv.org/pdf/2303.12570, https://arxiv.org/pdf/2403.16792). While the specific design in this paper appears new, the contribution is not clearly stated."
> - “The high-level idea of generating hypothetical scenarios or internal monologue before the final response has been explored before: https://arxiv.org/abs/2311.07445, https://arxiv.org/abs/2405.06373”
> - “The techniques used in this paper were not very novel (more specific prompts, LLM as a judge for evaluation, role-playing, more in depth personas). The idea is pretty generic (similar to many existing papers such as https://aclanthology.org/2023.acl-long.468.pdf, https://arxiv.org/abs/2402.10946)”
> - “The work is not novel and the experiment is not good enough. Many similar works already exist:  https://arxiv.org/abs/2309.07034 https://arxiv.org/abs/2311.09730”
> - “the methodological contribution is limited since the CSPP does not provide insight in prompt design beyond the scope of existing work.  [1] https://arxiv.org/abs/2104.10343 [2] https://arxiv.org/abs/2302.09664 [3] https://arxiv.org/pdf/2402.06782”
> - “The proposed method is very similar to prompting methods in relevant literature along the line of self criticism/self correction (e.g. Wang et al., 24, https://arxiv.org/abs/2305.13733).”
> - “It is not very novel because there are already papers that use translation to augment MCQA (https://arxiv.org/pdf/2012.05958)”
> - “The idea of generating code based on a plan and previous generation has been explored in prior work: https://arxiv.org/abs/2310.03302, https://arxiv.org/abs/2410.07095. This work explores prompting techniques that are used in many prior studies, with trivial variations.”

---

> ### Author Response · Authors · 2025-11-29
> **Response to Reviewer qsa4 (Part 2)**
>
> 2. *Would results differ with newer models (e.g., GPT-4o)?*
>
> While re-running the same human expert evaluation based on the execution outcomes would be prohibitively expensive and slow to carry out during a rebuttal, we provide some rough signals via LLM-judges as a proxy. Specifically, we generate 120 ideas for 4 different topics (30 ideas for each) following the same ideation agent scaffold used in our work, but with newer backbone models: Claude-4.5-Sonnet, GPT-5, and Gemini 3.0 Pro. We then use GPT-5 and Claude-4.5-Sonnet as judges to compute win rates of these newer models against the Claude-3.5-Sonnet baseline and report the average win rates as judged by the two judge models.
>
> We report the win rates below:
>
> | Model              | Novelty | Excitement | Effectiveness | Overall |
> |--------------------|---------|------------|----------------|---------|
> | Claude-4.5-Sonnet | 77.9%   | 67.1%      | 74.2%          | 71.9%   |
> | GPT-5             | 75.0%   | 66.7%      | 73.8%          | 74.2%   |
> | Gemini 3 Pro      | 79.6%   | 76.3%      | 80.4%          | 81.7%   |
>
> The topics we used for sampling the ideas include:
> - Coding agents: novel prompting or agent scaffolding methods for large language models to improve code generation and automated software engineering
> - Scalable oversight: empowering humans to oversee the output of an AI system, especially on tasks where the AI system outperforms humans
> - Synthetic data: novel methods to construct synthetic data to pretrain or finetune LLMs
> - Idea generation: novel methods to improve LLMs for idea generation
>
> Note that LLM-judge is a very noisy proxy, and we do see some moderate improvement in these newer models as ideators (average win rates above 50% against Claude-3.5-Sonnet for all metrics). For a more reliable and rigorous evaluation of newer ideation models, future works could adopt similar evaluation methods as our human study to test these newer models.
>
> 3. *Could a predicted “feasibility” score reduce the gap?*
>
> That’s a great question. One possibility here is to train a model to predict the effectiveness of an idea based only on the natural language idea itself. Then, we could potentially use this predictor as a reward model in an RL loop to train the ideator model to generate more effective ideas, or use the predictor to guide some sort of iterative revision or search. In order for this approach to succeed, we would need a reliable predictor model that can accurately predict the effectiveness (e.g., benchmark scores) of ideas.
>
> One recent work has explored this exact idea: “Predicting Empirical AI Research Outcomes
> with Language Models” (NeurIPS’2025). They scraped 6000 data points to finetune GPT-4.1 in a binary setting (i.e., to predict whether the idea would be more effective than a baseline on the given benchmarks). The best performance they can get from this finetuned model is merely 64.4% on this simplified binary setup (where the random baseline is 50%), which would be far too noisy to be used as a reward model in an RL or iterative feedback loop to improve the ideator model. Therefore, we believe more work is needed to build accurate idea effectiveness predictors before we can use them to improve the underlying idea generation model.
>
> 4, *According to Table 3, AI idea executors are more experienced than those for Human. Would this affect the results?*
>
> We did random idea assignment (within the executor’s preferred topics) to control for the expertise level. We conducted two-sided t-tests to compare executors from the AI condition and the Human condition:
>
> | Metric                          | p-value | Significant Difference |
> |---------------------------------|---------|-------------------------|
> | Execution Time                  | 0.0709  | No                      |
> | Executor Familiarity with Topic | 0.0769  | No                      |
> | Citation  Count                  | 0.6575  | No                      |
> | Paper Count                        | 0.9004  | No                      |
>
> Therefore, the expertise level between the two conditions is not statistically different.

---

> ### Author Response · Authors · 2025-11-29
> **Response to Reviewer qsa4 (Part 3)**
>
> 5. *Reviewer consistency is reported only in aggregate; topic-level calibration is missing.*
>
> We report the detailed reviewer consistency for all executed projects below:
>
> Across all topics:
>
> | Metric          | Consistency |
> |----------------------|-------------|
> | overall score        | 0.7050      |
> | novelty score        | 0.6700      |
> | excitement score     | 0.6825      |
> | soundness score      | 0.5900      |
> | effectiveness score  | 0.8425      |
>
>
>
> Topic-level breakdown:
>
> - Coding (n=6 ideas):
>
> | Metric         | Consistency |
> |----------------------|-------------|
> | overall score        | 0.9111      |
> | novelty score        | 0.5778      |
> | excitement score     | 0.8556      |
> | soundness score      | 0.4778      |
> | effectiveness score  | 0.6000      |
>
>
> - Factuality (n=10 ideas):
>
> | Metric          | Consistency |
> |---------------------|-------------|
> | overall score       | 0.7500      |
> | novelty score       | 0.6600      |
> | excitement score    | 0.6100      |
> | soundness score     | 0.7300      |
> | effectiveness score | 0.8800      |
>
>
> - Multilingual (n=7 ideas):
>
> | Metric         | Consistency |
> |---------------------|-------------|
> | overall score       | 0.7037      |
> | novelty score       | 0.8241      |
> | excitement score    | 0.7407      |
> | soundness score     | 0.5000      |
> | effectiveness score | 0.8796      |
>
>
> - Safety (n=6 ideas):
>
> | Metric          | Consistency |
> |---------------------|-------------|
> | overall score       | 0.5256      |
> | novelty score       | 0.5641      |
> | excitement score    | 0.4872      |
> | soundness score     | 0.5385      |
> | effectiveness score | 0.8205      |
>
>
> - Uncertainty (n=5 ideas):
>
>  | Metric          | Consistency |
> |---------------------|-------------|
> | overall score       | 0.7500      |
> | novelty score       | 0.6500      |
> | excitement score    | 0.7125      |
> | soundness score     | 0.0875      |
> | effectiveness score | 0.7500      |
>
>
> - Bias (n=7 ideas):
>
>  | Metric          | Consistency |
> |---------------------|-------------|
> | overall score       | 0.2188      |
> | novelty score       | 0.4583      |
> | excitement score    | 0.3125      |
> | soundness score     | 0.4271      |
> | effectiveness score | 0.5417      |
>
>
> Note that the reviewer consistency is generally high except for the Bias topic, potentially because the judgment of these projects tends to be more subjective than other topics. Also, we omitted the Math topic in the breakdown because it only has two ideas, and the consistency metric wouldn’t be reliable when the sample size is so small.
>
> 6. *Overall technical contribution is not significant.*
>
> Our work is the first large-scale human study to rigorously examine the effectiveness of AI-generated research ideas, and we are the first to demonstrate the existence of the ideation-execution gap of LLM-generated research ideas, which highlights the importance of considering empirical effectiveness in the evaluation process of research ideas and establishes the best practices for future work on evaluating AI-generated research ideas.
>
> Thus, despite not being a modeling paper, we believe our work makes important contributions towards the evaluation practices of an important research problem, and shares important insights on the limitations of LLM-generated ideas, all within the scope of this conference. In fact, multiple prior evaluation and analysis papers on LLMs for research ideation were published at recent ML conferences, such as “Can LLMs Generate Novel Research Ideas? A Large-Scale Human Study with 100+ NLP Researchers” (ICLR’2025), “MOOSE-Chem: Large Language Models for Rediscovering Unseen Chemistry Scientific Hypotheses” (ICLR’2025),  “ResearchTown: Simulator of Human Research Community” (ICML’2025), and “DiscoveryBench: Towards Data-Driven Discovery with Large Language Models” (ICLR’2025).

---

### Official Review · Reviewer_4Uap · 2025-11-01

**Soundness:** 3
**Presentation:** 3
**Contribution:** 2
**Rating:** 4
**Confidence:** 4

**Summary:**

This paper studies whether research ideas generated by large language models (LLMs) can lead to useful research outcomes once they are actually executed. Previous work has shown that LLMs are often able to propose ideas that look more novel or interesting than those written by human researchers. However, these earlier evaluations mostly stopped at the idea stage and did not test whether such ideas can be successfully turned into working research projects.
To address this gap, the authors conduct a controlled experiment involving 43 experienced NLP researchers. Each participant was randomly assigned either a human-written idea or an idea generated by the Claude 3.5-Sonnet model. They then spent around 100 hours implementing the idea and wrote a short 4-page report describing the experiments and results. All these completed projects were blindly reviewed by independent experts using consistent criteria, including novelty, excitement, effectiveness, and overall quality.
The results show a clear pattern: although LLM-generated ideas initially received higher scores at the ideation stage, their performance dropped significantly after execution. Human ideas, on the other hand, maintained roughly the same level of quality before and after execution. The authors refer to this difference as the ideation–execution gap. Further analysis indicates that the gap is not caused by poor implementation—since participants made only small adjustments to datasets or metrics—but rather by the lower practical feasibility and methodological soundness of the AI-generated ideas.

**Strengths:**

1.	The paper tackles an important and timely issue: whether LLM-generated research ideas can actually lead to successful scientific outcomes once executed. This question extends prior work that only evaluated idea quality (Si et al., ICLR 2025), moving a step forward to test execution outcomes directly.
2.	The study involves a large number of highly qualified human experts, including experienced researchers with strong publication records. It also quantifies key aspects such as working hours (around 100 hours per project) and the number of modifications made to each idea, showing that participants made only minor adjustments without altering core methods. These details make the findings convincing.
3.	The paper combines quantitative comparisons of review scores with qualitative analyses of reviewer comments and modification patterns. This mixed approach allows for a balanced understanding of why AI-generated ideas underperform at the execution stage.

**Weaknesses:**

1.	While the research question is important, the paper mainly contributes through experimental design rather than technical or algorithmic advances. It does not propose new modeling approaches or analytical frameworks beyond the comparative study setup.
2.	The paper identifies the existence of the ideation–execution gap but provides limited theoretical or computational explanation for why LLM-generated ideas fail in execution. A more systematic analysis of idea structure or model behavior could have strengthened the conclusions and would be more helpful for guiding models to generate better and more feasible research ideas in the future.

**Questions:**

1.	Each executed project was limited to around 100 hours of work and a 4-page report. Could this relatively short duration constrain the depth and maturity of the resulting research? For instance, some AI-generated ideas might be more ambitious or technically demanding, and therefore require substantially more time or resources to reach comparable quality.
2.	Did the authors consider normalizing for idea complexity or estimated implementation difficulty when comparing execution outcomes? Without such adjustment, simpler human ideas may appear more successful simply because they are easier to realize within the given time and effort constraints.
3.	Would it be possible to introduce AI-assisted execution in future experiments, so that the implementation process is more aligned with the characteristics of AI-generated ideas?

---

> ### Author Response · Authors · 2025-11-29
> **Response to Reviewer 4Uap (Part 1)**
>
> Thank you for your insightful review. To address your questions and concerns:
>
> 1. *While the research question is important, the paper mainly contributes through experimental design rather than technical or algorithmic advances. It does not propose new modeling approaches or analytical frameworks beyond the comparative study setup.*
>
> Our work is the first large-scale human study to rigorously examine the effectiveness of AI-generated research ideas, and we are the first to demonstrate the existence of the ideation-execution gap of LLM-generated research ideas, which highlights the importance of considering empirical effectiveness in the evaluation process of research ideas and establishes the best practices for future work on evaluating AI-generated research ideas.
>
> Thus, despite not being a modeling paper, we believe our work makes important contributions towards the evaluation practices of an important research problem, and shares important insights on the limitations of LLM-generated ideas, all within the scope of this conference. In fact, multiple prior evaluation and analysis papers on LLMs for research ideation were published at recent ML conferences, such as “Can LLMs Generate Novel Research Ideas? A Large-Scale Human Study with 100+ NLP Researchers” (ICLR’2025), “MOOSE-Chem: Large Language Models for Rediscovering Unseen Chemistry Scientific Hypotheses” (ICLR’2025),  “ResearchTown: Simulator of Human Research Community” (ICML’2025), and “DiscoveryBench: Towards Data-Driven Discovery with Large Language Models” (ICLR’2025).
>
>
> 2. *The paper identifies the existence of the ideation–execution gap but provides limited theoretical or computational explanation for why LLM-generated ideas fail in execution. A more systematic analysis of idea structure or model behavior could have strengthened the conclusions and would be more helpful for guiding models to generate better and more feasible research ideas in the future.*
>
> This is a great point! We manually analyzed all the free-text reviews of both LLM and human ideas to categorize the common weaknesses.
>
> For human ideas, the most common criticisms are: 1) the idea is not novel enough, and 2) the method is not effective enough. These two are also common criticisms of LLM ideas.
>
> However, LLM ideas have some additional weaknesses that are disproportionately mentioned, including:
> - **Poorly designed experiments, such as inappropriate metrics, missing baselines, and experiments not support the claims.**
> Some actual reviews for the LLM ideas include: “not using the same metrics as other works to compare the efficacy of this method”, “with no comparison against prior work”, “many, many related baselines are missed from this paper for fair evaluations”, “some claims are flawed or under-substantiated. With more rigorous experiments and more comprehensive analysis it could reach an acceptance threshold”  “the baseline comparison is far from enough. Importing results from existing work, even without replication can be a good way to address this”, “the justification for the experimental setup is rather weak, and the evaluation is not well justified with respect to literature in the literary world”, “the lack of empirical validation for most of the design decisions makes the experimental setup a bit questionable”, “claims that are unsupported by the results shown”, “lacks comparison with previous work: the method is only compared with the simplest baselines despite on well-acknowledged benchmarks”, and “no comparison against prior work, there is a lack of depth in the analysis”
> - **Ideas are not well-motivated and lack theoretical grounding.** Some actual reviews for the LLM ideas include: “lacks any clear theoretical grounding”, “The paper lacks a more grounded reasoning”,  “Very weird method that seems fundamentally flawed”, and “there's a critical lack of motivation or reference to prior work”.
>
> We also included a quantitative categorization in Figure 3 of the paper. We believe these would be a good starting point for future work to try to improve LLMs for idea generation.

---

> ### Author Response · Authors · 2025-11-29
> **Response to Reviewer 4Uap (Part 2)**
>
> 3. *Each executed project was limited to around 100 hours of work and a 4-page report. Could this relatively short duration constrain the depth and maturity of the resulting research? For instance, some AI-generated ideas might be more ambitious or technically demanding, and therefore require substantially more time or resources to reach comparable quality.*
>
> We gave every participant a three-month window to finish the execution, which we believe is sufficient for the scope of our ideas based on our pilot study and the participants’ feedback. The fact that our participants only took around 100 hours of work is largely because we deliberately constrained the scope of all ideas to be prompting-based research, which makes the execution more feasible. We also provided sufficient compute credits to all participants to cover all of their experiments, and made sure all of them had enough time and resources to finish all the experiments mentioned in the assigned ideas. Therefore, we believe our evaluation faithfully reflects the quality of the underlying ideas. For cases where the execution time is low, it’s usually because the ideas are simpler, rather than the participants lacking time or resources. We will showcase some quantitative and qualitative evidence of this in our response to the next point.
>
>
> 4. *Did the authors consider normalizing for idea complexity or estimated implementation difficulty when comparing execution outcomes? Without such adjustment, simpler human ideas may appear more successful simply because they are easier to realize within the given time and effort constraints.*
>
> We controlled the scope of all ideas to be prompting-based research, and we generated the human and LLM ideas on the same set of NLP topics. Therefore, the topic complexity is controlled. In the blinded, completely randomized design of our experiment, all the causal effects are attributed to whether the idea came from AI or humans. Thus, the causal direction here is known - if there is a significant difference in execution time, it comes from differences in ideas between humans and AIs.
>
> There is a difference in the execution time between the two conditions, where human ideas have an average execution time of 112.6 hours, while AI ideas have an average execution time of 93.7 hours.
>
> The correlation between execution time and the overall score of the executed project is 0.34, indicating a moderate correlation between execution time and the average score of the executed project.
>
> We manually reviewed the projects with the lowest execution time, and they do tend to correspond to rather low complexity. We present the summaries of two example ideas that correspond to the lowest execution time below:
>
> - [20 hours; AI idea; overall score 3.25] Sociolinguistic Role-Play Prompting (SRP), a novel technique that frames language tasks as a form of social role-play. SRP works by constructing detailed prompts that specify not just the task, but also the social identities and relationships of the participants, the setting, and the social norms at play.
> - [33 hours; AI idea; overall score 2.50] Culturally-Grounded Chain-of-Thought (CG-CoT), a prompting technique that interleaves cultural context injection with step-by-step reasoning. For each reasoning step, the model is prompted to first recall relevant cultural knowledge, then apply this knowledge to the task at hand.
>
> We contrast these with two example projects with the longest execution time:
>
> - [156 hours; AI idea; overall score 5.00] Adaptive Contextual Pruning (ACP), which involves: (1) Maintaining a dynamic relevance score for each piece of context based on its usage in recent generations. (2) Periodically prompting the model to identify the most relevant context pieces for the current generation task. (3) Pruning less relevant context to maintain a focused, manageable context window. (4) Allowing the model to 'retrieve' previously pruned context if it becomes relevant again, prompted by keywords or themes in the current generation.
> - [140 hours; Human idea; overall score 4.00] Multilingual Knowledge Abstaining (MKA). The key steps include:  (1) Translate the given instruction from the target language into multiple auxiliary languages. (2) Autoregressively generate the response using the target LM on each of the auxiliary language instructions separately. (3) Translate the auxiliary language responses back to the target language, potentially also performing canonicalization. (4) Compute agreement level; abstain if the agreement is below a certain threshold tuned on a validation set.
>
> So, it is not the case that “human ideas may appear more successful simply because they are easier to realize within the given time and effort constraints”, but rather, the AI condition contains more simple ideas (as shown in the lower execution time and manual analysis).

---

> ### Author Response · Authors · 2025-11-29
> **Response to Reviewer 4Uap (Part 3)**
>
> 5. *Would it be possible to introduce AI-assisted execution in future experiments, so that the implementation process is more aligned with the characteristics of AI-generated ideas?*
>
> Yes. In fact, for this experiment, we didn’t ban the use of AI coding assistant tools, and it’s possible the participants are already doing so. We just asked our reviewers to check the quality of the submitted codebases and made sure they have high quality (3.58 out of 5 for both conditions).

---

### Official Review · Reviewer_P5TK · 2025-11-04

**Soundness:** 4
**Presentation:** 4
**Contribution:** 3
**Rating:** 6
**Confidence:** 5

**Summary:**

This paper investigates the "ideation-execution gap" by comparing research ideas generated by an LLM (Claude-3.5-Sonnet) with those from human experts through a rigorous Randomized Controlled Trial (RCT). 43 expert researchers executed the assigned NLP ideas (~100 hours each) and produced short papers, which were then blindly reviewed by 58 expert reviewers. The core finding is that despite LLM ideas scoring significantly higher in novelty and excitement at the ideation stage, their scores dropped significantly more than human ideas after execution (p<0.05) across all metrics (novelty, excitement, effectiveness, and overall). This reversal of rankings suggests that current LLMs struggle to generate ideas that are truly effective when subjected to the rigors of practical research execution.

**Strengths:**

1. The use of a large-scale Randomized Controlled Trial (RCT) with blinding for both execution participants and reviewers is highly commendable and provides a strong foundation for the causal conclusions.
2. The study mandated significant effort, with 43 expert participants spending an average of over 100 hours each to execute the ideas, lending credibility to the post-execution evaluation outcomes.
3. The direct comparison of pre- and post-execution scores explicitly quantifies the difference in "drop" between AI-generated and human-generated ideas, providing a novel and important metric for evaluating AI ideation capabilities.

**Weaknesses:**

1. While the drop in scores is statistically significant, the direct comparison of the final average execution scores between human and AI ideas was not statistically significant when treating each idea as an independent data point ($N=43$)8. This tempers the claim of human ideas being superior post-execution.
2. Execution participants for human ideas spent significantly more time (mean 112.6 hr) compared to AI ideas (mean 93.7 hr). Although potentially explained by sample size, this difference could be a confounding variable related to the complexity or inherent feasibility of the assigned ideas.
3. The study used a single, albeit state-of-the-art at the time, LLM (Claude-3.5-Sonnet) and generation method. It is unclear how different LLMs or more advanced agentic idea generation frameworks (e.g., iterative feedback) would impact the "execution gap".

**Questions:**

The analysis points out that execution reviewers focus more on empirical performance and rigor (e.g., missing baselines, insufficient analysis). Can the authors provide a fine-grained analysis of which specific types of weaknesses (e.g., technical flaws, inadequate experiment design, poor empirical results) were disproportionately mentioned in the low-scoring AI idea reviews compared to the low-scoring human idea reviews? This would help precisely characterize the nature of the LLM's ideation failure.
The disparity in mean execution time (112.6 hrs for Human vs. 93.7 hrs for AI) is a potential confounder. Did the projects with lower execution time generally receive lower reviewer scores, and was the difference in execution time due to AI ideas being inherently simpler, or were they less fully explored by the executors?

---

> ### Author Response · Authors · 2025-11-29
> **Response to Reviewer P5TK (Part 1)**
>
> Thank you for your insightful review. To address your questions and concerns:
>
> 1. *While the drop in scores is statistically significant, the direct comparison of the final average execution scores between human and AI ideas was not statistically significant when treating each idea as an independent data point. This tempers the claim of human ideas being superior post-execution.*
>
> Yes, we have acknowledged this in the paper and faithfully claimed in Sec 4.1 that “we do not have sufficient statistical power to directly confirm our pre-registered hypothesis that human ideas differ significantly from AI ideas in the execution scores”. Regardless, our main finding on the ideation-execution gap is statistically significant across the human and AI conditions, as shown in Sec 4.2, and this is the core conclusion of the paper.
>
> 2. *Can the authors provide a fine-grained analysis of which specific types of weaknesses (e.g., technical flaws, inadequate experiment design, poor empirical results) were disproportionately mentioned in the low-scoring AI idea reviews compared to the low-scoring human idea reviews?*
>
> This is a great point! We manually analyzed all the free-text reviews of both LLM and human ideas to categorize the common weaknesses.
>
> In human ideas, the most common criticisms are: 1) the idea is not novel enough, and 2) the method is not effective enough. These two are also common criticisms of LLM ideas.
>
> However, LLM ideas have some additional weaknesses that are disproportionately mentioned, including:
> - **Poorly designed experiments, such as inappropriate metrics, missing baselines, and experiments not support the claims.**
> Some actual reviews for the LLM ideas include: “not using the same metrics as other works to compare the efficacy of this method”, “with no comparison against prior work”, “many, many related baselines are missed from this paper for fair evaluations”, “some claims are flawed or under-substantiated. With more rigorous experiments and more comprehensive analysis it could reach an acceptance threshold”  “the baseline comparison is far from enough. Importing results from existing work, even without replication can be a good way to address this”, “the justification for the experimental setup is rather weak, and the evaluation is not well justified with respect to literature in the literary world”, “the lack of empirical validation for most of the design decisions makes the experimental setup a bit questionable”, “claims that are unsupported by the results shown”, “lacks comparison with previous work: the method is only compared with the simplest baselines despite on well-acknowledged benchmarks”, and “no comparison against prior work, there is a lack of depth in the analysis”
> - **Ideas are not well-motivated and lack theoretical grounding.** Some actual reviews for the LLM ideas include: “lacks any clear theoretical grounding”, “The paper lacks a more grounded reasoning”,  “Very weird method that seems fundamentally flawed”, and “there's a critical lack of motivation or reference to prior work”.
>
> We also included a quantitative categorization in Figure 3 of the paper.

---

> ### Author Response · Authors · 2025-11-29
> **Response to Reviewer P5TK (Part 2)**
>
> 3. *The disparity in mean execution time (112.6 hrs for Human vs. 93.7 hrs for AI) is a potential confounder. Did the projects with lower execution time generally receive lower reviewer scores, and was the difference in execution time due to AI ideas being inherently simpler, or were they less fully explored by the executors?*
>
> We agree that there is a difference in the execution time between the two conditions, although this difference is not statistically significant (p=0.07 under a two-sided t-test).
>
> The correlation between execution time and the overall score of the executed project is 0.34, indicating only a moderate correlation between execution time and the average score of the executed project.
>
> We manually reviewed the projects with the lowest execution time, and they do tend to correspond to rather low complexity. We present the summaries of two example ideas that correspond to the lowest execution time below:
>
> - [20 hours; AI idea; overall score 3.25] Sociolinguistic Role-Play Prompting (SRP), a novel technique that frames language tasks as a form of social role-play. SRP works by constructing detailed prompts that specify not just the task, but also the social identities and relationships of the participants, the setting, and the social norms at play.
> - [33 hours; AI idea; overall score 2.50] Culturally-Grounded Chain-of-Thought (CG-CoT), a prompting technique that interleaves cultural context injection with step-by-step reasoning. For each reasoning step, the model is prompted to first recall relevant cultural knowledge, then apply this knowledge to the task at hand.
>
> We contrast these with two example projects with the longest execution time:
>
> - [156 hours; AI idea; overall score 5.00] Adaptive Contextual Pruning (ACP), which involves: (1) Maintaining a dynamic relevance score for each piece of context based on its usage in recent generations. (2) Periodically prompting the model to identify the most relevant context pieces for the current generation task. (3) Pruning less relevant context to maintain a focused, manageable context window. (4) Allowing the model to 'retrieve' previously pruned context if it becomes relevant again, prompted by keywords or themes in the current generation.
> - [140 hours; Human idea; overall score 4.00]Multilingual Knowledge Abstaining (MKA). The key steps include:  (1) Translate the given instruction from the target language into multiple auxiliary languages. (2) Autoregressively generate the response using the target LM on each of the auxiliary language instructions separately. (3) Translate the auxiliary language responses back to the target language, potentially also performing canonicalization. (4) Compute agreement level; abstain if the agreement is below a certain threshold tuned on a validation set.
>
> So, to answer your questions: AI ideas do have slightly lower execution time; across both conditions, there is a positive correlation between the execution time and average score; and upon our manual inspection of the ideas, ideas with the lowest execution time do tend to be more straightforward ideas that likely explain the lower execution time. In the blinded, completely randomized design of our experiment, all the causal effects are attributed to whether the idea came from AI or humans. Thus, the causal direction here is known - if there is a significant difference in execution time, it comes from differences in ideas between humans and AIs.

---

### Meta-Review · Area_Chair_TcSe · 2025-12-28

**Summary:**

The paper investigates the "ideation-execution gap" by conducting a large-scale randomized controlled trial where expert researchers execute LLM-generated (Claude 3.5-Sonnet) and human expert research ideas in NLP topics, followed by blind reviews of the resulting papers. It finds that LLM ideas initially score higher in novelty and excitement but experience a significantly greater drop in scores (novelty, excitement, effectiveness, overall) after execution compared to human ideas, highlighting limitations in LLM-generated research viability.

All reviewer concerns were addressed, such as execution time disparities, potential training data contamination, impacts of newer models (e.g., GPT-4), and others. The motivation of the work is strong, and the extensive experiments and analyses adequately support the paper's arguments. Thus, I recommend accepting this paper.

**Reviewer Concerns:**

All concerns were addressed.

**Reviewer Scores:**

No reviewer would change the score.

---

### Decision · Program_Chairs · 2026-01-26

Accept (Poster)